

# A social information sensitive model for conversational recommender systems

Abdulaziz Mohammed[1], Mingwei Zhang[1], Gehad Abdullah Amran[2], Husam M. Alawadh[3], Ruizhe Wang[1], Amerah Alabrah[4] and Ali A. Al-Bakhrani[5]

[1] College of Software Engineering, Northeastern University, Shenyang, China
[2] Department of Management Science and Engineering, Dalian University of Technology, Dalian, China
[3] Department of English Language, College of Language Sciences, King Saud University, Riyadh, Saudi Arabia
[4] Department of Information Systems, College of Computer and Information Science, King Saud University, Riyadh, Saudi Arabia
[5] College of Software Engineering, Dalian University of Technology, Dalian, China

Corresponding authors
Mingwei Zhang,
zhangmw@swc.neu.edu.cn
Gehad Abdullah Amran,
jehad.westran@gmail.com

## ABSTRACT

Conversational recommender systems (CRS) facilitate natural language interactions for more effective item suggestions. While these systems show promise, they face challenges in effectively utilizing and integrating informative data with conversation history through semantic fusion. In this study we present an innovative framework for extracting social information from conversational datasets by inferring ratings and constructing user-item interaction and user-user relationship graphs. We introduce a social information sensitive semantic fusion (SISSF) method that employs contrastive learning (CL) to bridge the semantic gap between generated social information and conversation history. We evaluated the framework on two public datasets (ReDial and INSPIRED) using both automatic and human evaluation metrics. Our SISSF framework demonstrated significant improvements over baseline models across all metrics. For the ReDial dataset, SISSF achieved superior performance in recommendation tasks (R@1: 0.062, R@50: 0.437) and conversational quality metrics (Distinct-2: 4.223, Distinct-3: 5.595, Distinct-4: 6.155). Human evaluation showed marked improvement in both fluency (1.81) and informativeness (1.63). We observed similar performance gains on the INSPIRED dataset, with notable improvements in recommendation accuracy (R@1: 0.046, R@10: 0.129, R@50: 0.269) and response diversity (Distinct-2: 2.061, Distinct-3: 4.293, Distinct-4: 6.242). The experimental results consistently validate the effectiveness of our approach in both recommendation and conversational tasks. These findings suggest that incorporating social context through CL can significantly improve the personalization and relevance of recommendations in conversational systems.

## INTRODUCTION

Conversational Recommender Systems (CRS) enhance item recommendations through natural language interactions (*Jannach et al., 2021*). Studies have shown that integrating external data sources with conversational data through semantic fusion provides opportunities to improve CRS models (*Zhou et al., 2020a*, *2022*; *Pugazhenthi & Liang, 2022*). Semantic fusion unifies different types of data representations into a single space, which combines all the characteristics of the involved data. Incorporating various data sources enables CRS to capture user preferences more effectively, which leads to better recommendations (*Zhou et al., 2020a*; *Chen et al., 2019*). Additionally, semantic fusion can expand the model's vocabulary by providing more of a general insight into natural language patterns (*Pugazhenthi & Liang, 2022*). Moreover, it integrates user interactions for more personalized recommendations (*Kannout et al., 2024*). However, while previous studies (*Chen et al., 2019*; *Zhou et al., 2020a*, *2022*; *Pugazhenthi & Liang, 2022*) have provided valuable insights into the effects of semantic fusion in CRS, it seems that the potential impact of social information that is derived from the relationships amongst users and the interactions of users and items has not been fully explored.

There are two main challenges with leveraging social information in CRS. Firstly, popular conversational datasets, such as the ReDial (*Li et al., 2018*) and the INSPIRED (*Hayati et al., 2020*) datasets lack explicit structural representations of social relationships and user interactions, specifically in the form of user-item interaction and user-user relationship graphs. This absence limits the system's ability to utilise valuable social dynamics. Secondly, a significant semantic gap exists between social information and conversation history, which makes it challenging to effectively integrate user preferences expressed in natural language with patterns derived from social relationships. To address these limitations, we propose SISSF CRS, a novel framework that enhances CRS through two primary mechanisms. Firstly, it inductively extracts social information from conversational datasets by constructing user-item interaction and user-user relationship graphs derived from explicit user interactions and inferred ratings through neural graph collaborative filtering (NGCF) (*Wang et al., 2019*). Secondly, it implements an innovative semantic fusion approach utilising CL to bridge the semantic gap between social information and conversation history.

To extract social information from conversational datasets, we first utilize NGCF (*Wang et al., 2019*) to infer item ratings based on user interactions. These ratings are then used to construct a user–item interaction graph *via* relational graph convolution network (R-GCN) (*Schlichtkrull et al., 2018*). Inspired by the work of *Hayati et al. (2020)*, which builds a social CRS using social science research that considers personal opinions and encouraging communication, we build the social relationship graph using graph transformer operator (GTO) (*Shi et al., 2021*). We are further motivated by *Fan et al. (2019)*, *Tran, Snášel & Nguyen (2023)* to build a social recommender system that defines user embeddings by considering user-item interaction and user-user relationship graphs; therefore, we aggregate the constructed graphs and obtain the embedding for all users who have interacted with each item in the conversation history from both graphs separately.

Next, we apply pooling to generate a collective representation of user interactions and social profiles. This step is followed by concatenating both profiles and performing mean pooling to obtain enhanced user representations. Finally, by applying self-attention to these enhanced representations across all items in the conversation, we form a compact representation of social information. This representation captures the underlying patterns and relationships among users, which helps us understand user preferences and interests within the context.

We utilize contrastive learning (CL) (*Hu et al., 2023*; *Si, Jia & Jiang, 2024*) to integrate extracted social information with conversational history. CL has demonstrated strong capabilities in unifying distinct data representations, particularly those involving textual data. It has been successfully applied to combine text with images (*Zhang et al., 2021*), video segments (*Nan et al., 2021*), and structured and unstructured data (*Zhou et al., 2020a*, *2022*, *2020b*). CL functions by contrasting positive pairs—composed of a conversation history embedding and the corresponding social information—with negative pairs, where the conversation history embedding is matched with social information from a different sample within the batch. This contrastive mechanism helps align user preferences expressed in conversations with the influence of social connections and shared interests inferred from social data. As a result, it produces a richer and more accurate representation of user preferences.

Figure 1 illustrates the clear advantage of incorporating social information directly from conversations into CRS. Traditional CRS methods often rely on external sources such as entity relationship graphs or user reviews to enhance contextual understanding. However, these sources are frequently sparse and incomplete (*Zhang et al., 2023*), which can lead to a trial-and-error approach in refining recommendations. In contrast, the social information sensitive semantic fusion (SISSF) CRS extracts social dynamics directly from conversational data by constructing two types of graphs: a user-item interaction graph, which captures individual preferences and interactions, and a social relationship graph, which captures user connections. This dual-graph approach enables more precise and contextually relevant recommendations. For instance, if Alice is searching for a science fiction movie, SISSF CRS identifies her connection to Bob *via* the social relationship graph. It also recognizes Bob's strong preference for 'Interstellar (2014)' through the user-item interaction graph. Consequently, the system prioritizes 'Interstellar (2014)' as the top recommendation for Alice, whereas a traditional CRS would rely solely on a general entity relation graph and suggest options based only on genre.

Our work makes several significant contributions to the field:

- Building on previous social science research on mutual engagement and interest, we propose a novel methodology for extracting implicit social information from conversational datasets. Although these datasets do not explicitly contain detailed user interactions or relationships, our approach inducts social information from a conversations. As far as we know, this is the first attempt to capture such implicit dynamics in this manner.

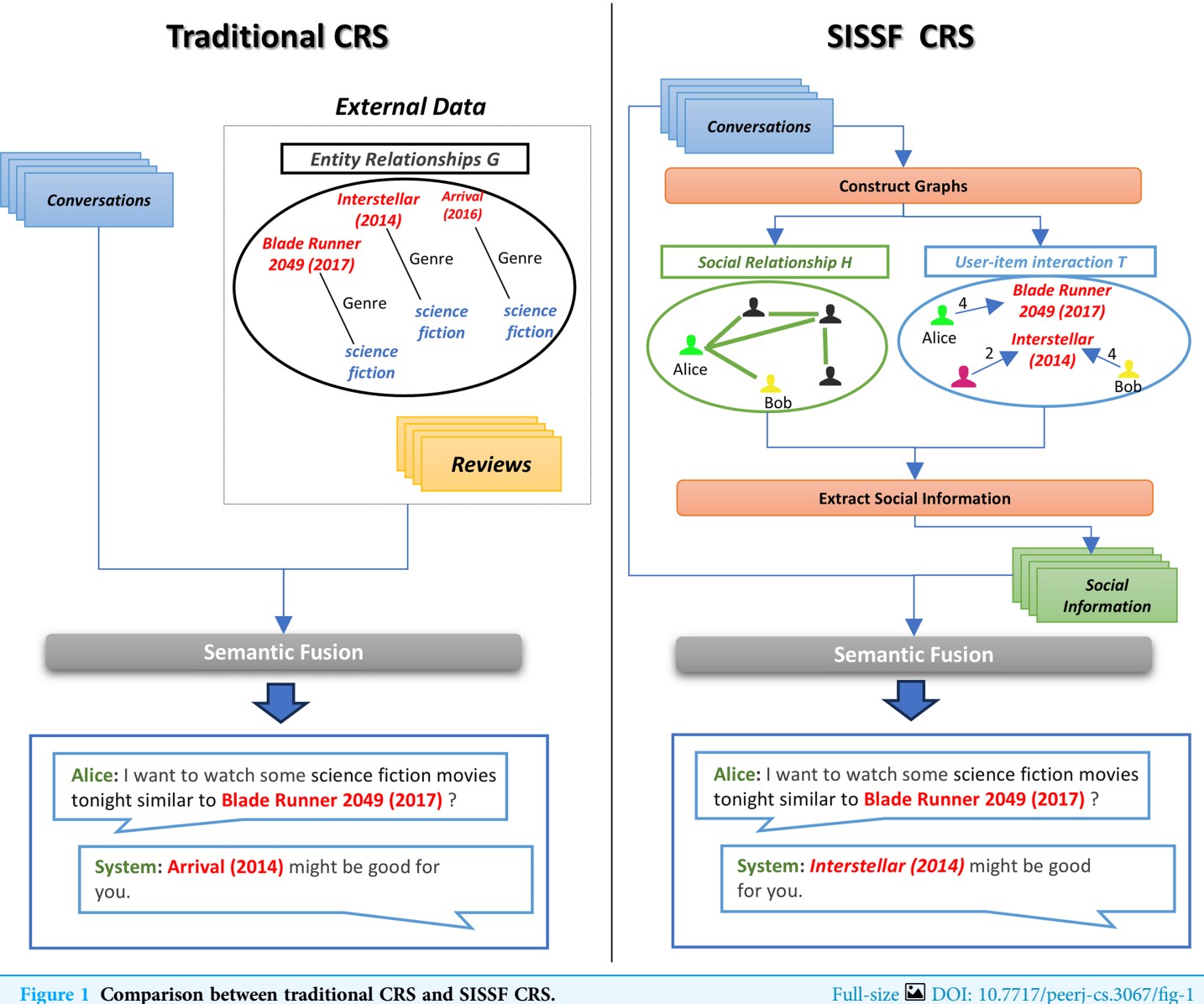

**Figure 1 Comparison between traditional CRS and SISSF CRS.**

- We introduce a novel semantic fusion approach that leverages CL to align social information extracted with conversation history. This critical integration step is essential for generating personalized and contextually relevant recommendations.
- Through extensive experimentation on two well-known public datasets, we demonstrate that our approach achieves statistically significant improvements compared to existing baseline models in both recommendation accuracy and response generation quality and diversity.
- We conducted extensive ablation studies comparing the SISSF CRS model with a variant that omits the semantic fusion process. The studies demonstrated statistically significant improvements in personalized recommendations and performance when semantic

fusion is employed. This validates our approach and confirms that the performance improvement is due to the effective integration of social dynamics with conversational history rather than simply overfitting with additional data.

The rest of this article is organized as follows: "Related Work" reviews related work in CRS including reinforcement learning (RL) based CRS, deep reinforcement learning (DRL) based CRS, deep learning (DL) based CRS, and CL. "Preliminaries" presents preliminary concept, notation, and task definition. "Approach" details our proposed methodology. "Experiment" presents experimental results and analysis. Finally, "Conclusion and Future Work" concludes the article and discusses future work and limitations.

# RELATED WORK

## Recommender system

Recommender systems are essential in providing personalized suggestions within commercial platforms, employing a variety of methodologies to cater to user preferences. However, traditional approaches, such as collaborative filtering (CF) (*Elahi, Ricci & Rubens, 2016*), rely on accumulated historical data, while Content-based systems (*Semeraro et al., 2009*) depend on item attributes and similarities. However, these conventional models often struggle to accommodate sudden shifts in user interests. CRS has emerged to overcome limitations (*Jannach et al., 2021*). CRS represents a specialized category that interacts with users through a dialogue interface, allowing users to specify desired items or attributes and make the model dynamically capture user interest. The system's goal is to comprehend and generate appropriate responses. CRS can be segmented into three groups based on managing interactions between the recommender and conversational components, as described in the subsequent sections.

## Reinforcement learning based CRS

In the field of CRS, RL has emerged as an esential technique for optimizing dialogue policies and enhancing user interactions. For instance, *Christakopoulou, Radlinski & Hofmann (2016)*, *Mahmood & Ricci (2009)* utilizes RL to refine the dialogue policy, determining subsequent actions at each dialogue turn. However, this approach relies on predefined reward functions, which can reduce adaptability to refine recommendations based on dynamic user needs. In contrast, *Chu, Wang & Wang (2023)* advocates for learning rewards intrinsically from user feedback instead of relying on predefined rewards. Similarly, *Zhang et al. (2020)*, where a contextual bandit algorithm is deployed to learn policies that select the best action. The data fusion method in *Li et al. (2021)* integrates items and attributes into a unified space, treating them as indistinct arms and applying Thompson Sampling for optimization. However, These techniques introduce challenges in accurately interpreting user feedback as they use predefined dialog templates to compose responses.

## DRL based CRS

In DRL approaches, *Lei et al. (2020)*, *Sun & Zhang (2018)*, *Sonie (2022)* introduce a deep neural network to represent the policy function. In *Lei et al. (2020)*, the policy network inputs the dialogue state and outputs action probabilities, training *via* the policy gradient method to maximize expected returns. Concurrently, *Sun & Zhang (2018)* situate DRL in the Action stage, formulating a dialogue policy that decides whether to inquire about attributes or suggest items, informed by the Estimation stage and conversation history. This article also presents a reward function that accounts for user feedback and dialogue success, employing policy gradient methods for optimal policy learning. These methodologies, however, present challenges in natural language understanding (NLU). The systems' reliance on optimal policy for interaction enhancement may compromise their ability to process complex, ambiguous, or diverse user requests or to deliver coherent, informative, and engaging responses.

## DL based CRS

Recent advancements in CRS have seen a shift from RL to DL methodologies for optimizing dialogue policies. Early DL based systems (*Li et al., 2018*; *Ghazvininejad et al., 2018*; *Liao et al., 2019*; *Patidar et al., 2018*; *Zhang et al., 2018*; *Zhou et al., 2020b*; *Wang et al., 2022*; *Zhou et al., 2020a*) employed supervised or unsupervised learning to encode user utterances and items, which the DL model processed to generate responses using various components and techniques. For instance, the seq2seq model (*Li et al., 2018*) is utilized for conversation generation and an autoencoder for recommendation, supplemented by a sentiment analysis unit to align the sentiments of both models. However, this approach could yield inaccurate results when entities are mentioned in user utterances without clear sentiment indicators. To address this, named entity recognition (NER) (*Ghazvininejad et al., 2018*), neural latent topic-based components (*Liao et al., 2019*), and slot-filling techniques (*Patidar et al., 2018*) were introduced to extract entities and construct a hierarchical category for the user's query. Despite these improvements, challenges persisted, particularly when entities were mentioned in a negation context or when capturing user feedback on preferences. A novel approach (*Zhang et al., 2018*) involved a search component that interacted with users to narrow down the item space, enhancing model efficiency. Further, a historical interaction augmented CRS (*Zhou et al., 2020b*) proposed to learn from both current and historical dialogues by merging embeddings. Prompt learning (PL) (*Wang et al., 2022*) emerged as a new method, using templates to guide pretrained language models (PLMs) in generating responses and recommendations. However, these techniques often struggle with the complexity and ambiguity of human language, such as sarcasm, irony, humor, and slang, which can be challenging to interpret correctly. To overcome these limitations, a more sophisticated approach (*Zhou et al., 2020a*, *2022*) is developed, providing user utterances with additional context by integrating multiple data sources. This method aimed to bridge the semantic gap between natural context data and structured data like knowledge graph (KG). Employing mutual information maximization (MIM) and a coarse-to-fine strategy. However, these approaches utilized spare and incomplete external data, which makes it

difficult to integrate highly relevant information into the current context. To better utilize the external data, variational reasoning over incomplete knowledge graphs for conversational recommendation (VRICR) (*Zhang et al., 2023*) uses a variational Bayesian approach that dynamically selects only the parts of the KG most relevant to the current dialogue. Although this leads to an improvement in performance over the previous methods, it still depends on a general-purpose graph that may not capture long-term user preferences as it lacks the fusion of social data, which leads to imprecise recommendations. Therefore, our innovative approach aims to surpass existing limitations by enriching the semantic fusion process with additional details that encapsulate user relationships and behaviors. Consequently, this approach narrows the recommended items space to the most related items based on the users' social circle.

### Application of CL

CL has emerged as a potent method for maximizing mutual information among data samples. By contrasting samples, CL acquires features that are both informative and discriminative, thus enhancing the representation of the data space. Recent studies have validated CL's efficacy in diverse data types, especially in NLU (*Chen et al., 2020*; *Zhang et al., 2021*; *Nan et al., 2021*; *Wu et al., 2020*; *Logeswaran & Lee, 2018*; *He et al., 2020*; *Bian et al., 2021*; *Hadsell, Chopra & LeCun, 2006*; *Hu et al., 2023*), and underscored its utility in complex applications like recommender systems (*Jing et al., 2023*). Specifically, CL has shown promising results in training disentangled datasets, which aim to improve the precision of personalized recommendations by considering the different user intents in interaction with items (*Ren et al., 2023*). Moreover, it has improved dimming dataset bias (*Zhou et al., 2023*) when applied in a causal framework using implicit preference satisfaction (IPS) and improved recommendation task for cross-domain sequential recommender (*Ye, Li & Yao, 2023*). Furthermore, to align separate bipartite graphs, CL achieves satisfactory outcomes when it is applied to improve the tagging system by making the representation of different graphs closer to a better understanding of user preferences (*Xu et al., 2023*). In CRS, CL applied to fuse the conversational history with external data sources (*Zhou et al., 2022*) to generate many rich responses powered by the resources available with the external data sources. In contrast, CL integrated external resources with the conversation history without explicitly mentioning entities (*Yang et al., 2023*). Our approach leverages CL to gather valuable information from a different data angle (*Hu et al., 2023*; *Si, Jia & Jiang, 2024*), such as social information extracted from the dataset itself. In doing so, it gains a deeper insight into what users prefer and provides recommendations that are both personalized and sensitive to the context of the user's needs.

## PRELIMINARIES

CRS integrates dialogue-based interactions with recommendation capabilities, allowing the system to infer user preferences dynamically. CRS models rely on conversational and recommendation components, but our approach extends this framework by incorporating

social relationship and user-item interaction graphs. This section outlines the key notations used in our system:

**Notation for conversation history.** In the domain of CRS, a conversation history (*Zhou et al., 2020a*, *2022*), denoted as $C$, represents an ordered collection of utterances $C = \{s_t\}_{t=1}^{n}$, where each $s_t$ corresponds to a turn in the conversation at time $t$. Each $s_t$ is further broken down into a sequence of words $s_t = \{w_j\}_{j=1}^{m}$ selected from a predefined vocabulary $V$. As the dialogue progresses, these utterances cumulatively form the conversation history.

**Notation for the user-item interaction bipartite graph.** Consider the sets $U = \{u_1, u_2, \ldots, u_n\}$ and $I = \{i_1, i_2, \ldots, i_m\}$ to represent users and items, respectively, where $n$ is the total count of users and $m$ is the total count of items. We define $R$ as the user-item rating matrix within $\mathbb{R}^{n \times m}$, also referred to as the user-item graph $T$. In this matrix, if user $u_i$ assigns a rating to item $i_j$, the rating is denoted by $r_{ij}$. If there is no rating from $u_i$ to $i_j$, it is indicated as $r_{ij} = 0$. The known rating $r_{ij}$ represents the opinion of user $u_i$ towards item $i_j$ that is a numerical representation of the user's level of preference regarding the item. Furthermore, we define $B(j)$ as the collection of users who have interacted with item $i_j$, and denote by $I_c$ the set of items involved in the conversation $C$.

**Notation for the user-user social homogeneous graph.** In ReDial (*Li et al., 2018*) and INSPIRED (*Hayati et al., 2020*), users remain the same throughout conversations but take on different roles depending on the interaction. A seeker requests recommendations, while a recommender provides them. These roles alternate dynamically based on the conversations. Let $K \in \mathbb{R}^{n \times n}$ define the matrix of user relationships where $k_{ij} = 1$ if there is a social relationship between $u_j$ and $u_i$, and $k_{ij} = 0$ otherwise, which represents the user-user social graph $H$. Following *Hayati et al. (2020)*, we form a connection if two users share at least one liked item—an item both have explicitly stated as enjoyable during at least one conversation. Additionally, since the recommender user plays a role in shaping the seekers' opinion, we also establish connections among seekers who have engaged with the same recommender, following the previously defined criteria.

**Task definition.** At the $t$-th turn, given the user-item interaction graph $T$, the user-user social graph $H$, and the conversational history $C$. The task is to (1) generate appropriate response $s_{t+1}$ and (2) recommend personalized items $I_{t+1}$.

## APPROACH

In this section, we systematically outline the development process of our SISSF CRS. Firstly, we explore the encoding of various data types used by our framework, including user-user relationship graph, user-item interaction graph, social information, and conversation history. These encoded representations are then fused through CL to establish semantic alignment across different sources of data. Finally, we fine-tune these representations to optimize their performance for recommendation and conversational tasks. Our novel framework is illustrated in Fig. 2.

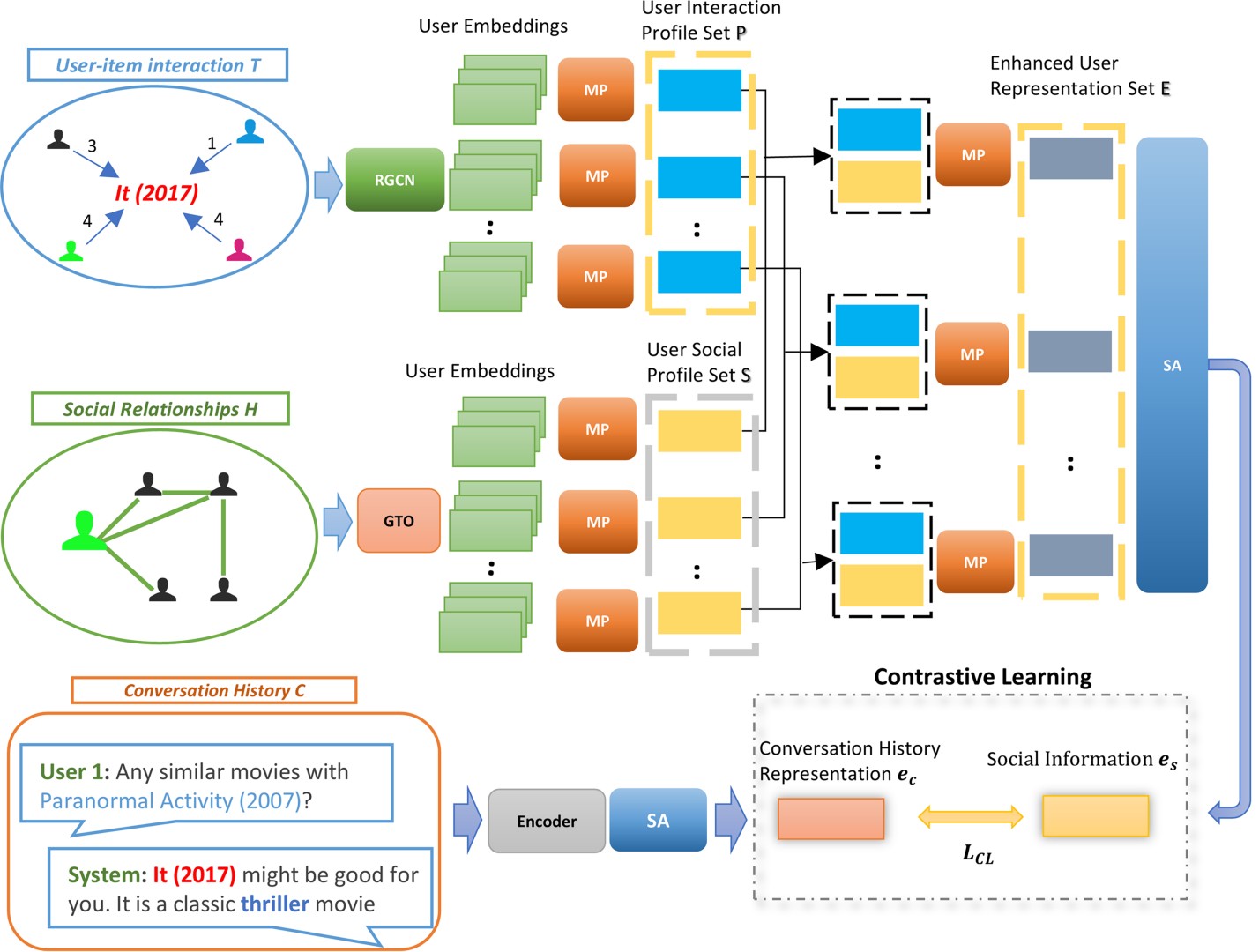

**Figure 2** Our approach to the sensitive information semantic fusion method involves leveraging CL to align the social information $e_s$ with the conversation history representation $e_c$. This process incorporates the social context by considering enhanced user representation set E of all items during the semantic fusion. Initially, the user-item interaction graph $T$ and social relationship graph $H$ are aggregated, followed by extracting the user interaction profile $P_i$ and the user social profile $S_i$, respectively, for each item $i_j$ in $C$ by applying MP to combine the user embeddings for each $i_j$. Then, we stack the representations for each $i_j$ separately and apply MP to obtain enhanced user representation $E_i$. Then, SA layer is used on E to generate social information representation $e_s$. Finally, these representations are fused with the encoded representation of $C$ through CL.

### Encoding multi-form data

Encoding transforms data from its original form into another form that deep learning models can process. This allows the model to interpret and preserve the semantic meaning while aligning the model's architecture with the data concerning its nature. Once the data is encoded, the representations are ready for applying CL to enhance their alignment and integration.

### Encoding of conversation history

Encoding the conversation history $C$ involves concatenating the utterances that constitute $C$ in the same order in which they appear, following the approach used in previous works (*Zhou et al., 2020a, 2022*). Specifically, we merge the sequence of utterances $\{s_t\}_{t=1}^{n}$ into a single paragraph $P$. Next, we encode $P$ with the help of a standard transformer encoder (*Vaswani et al., 2017*) as defined in Eq. (1) to obtain $h_c$ that is the contextual embedding of $C$.

$$h_c = \text{Encoder}(P), \tag{1}$$

To weigh each word in $h_c$ according to its contribution, we apply self-attention (SA) given by Eq. (2) to get conversation history representation $e_c$.

$$e_c = h_c \cdot \text{softmax}\big(\mathbf{b}^\top \cdot \tanh(W_{sa} \cdot h_c)\big), \tag{2}$$

where $W_{sa}$ and $b$ are the learnable parameters.

### Encoding of user-user relationship graph

In the social relationship graph $H$, nodes correspond to users in the set $U$, and relationships between users are represented as edges in the form of $\langle u_i, u_j \rangle$, where $u_i, u_j \in U$. To accurately capture and encode the significance of these relationships, we employ the GTO introduced in *Shi et al. (2021)* (see the ablation study section for further details). Specifically, we calculate a distinct vector for each user $u$ at layer $(l+1)$ as identified by Eqs. (3) and (4).

$$n_u^{(l+1)} = W_1^{(l)} n_u^{(l)} + \sum_{u' \in N(u)} \alpha_{u,u'}^{(l)} W_2^{(l)} n_{u'}^{(l)}, \tag{3}$$

$$\alpha_{u,u'}^{(l)} = \text{softmax}\left( \frac{\left(W_3^{(l)} n_u^{(l)}\right)^\top \left(W_4^{(l)} n_{u'}^{(l)}\right)}{\sqrt{d}} \right), \tag{4}$$

where $\alpha_{u,u'}^{(l)}$ denotes the attention coefficient. softmax is an activation function, $d$ represents the dimension of the feature vectors, $N(u)$ represents all neighbor nodes connected to the node $u$, the terms $W_1^{(l)}$, $W_2^{(l)}$, $W_3^{(l)}$, and $W_4^{(l)}$ represent learnable parameters, and $n_{u'}^{(l)}$ is the node $u'$ representation from the preceding layer $l$.

### Encoding of user-item interaction graph

In the user-item graph $T$, ratings provide important information about the strength of interaction between users and items. These interactions are denoted by triplets in the form $\langle u_i, r_k, i_j \rangle$, where $u_i \in U$, $i_j \in I$, and $r \in R$, and $r$ represents a specific type of relationship drawn from the set $R = \{0, 1, 2, 3, 4, 5\}$ that connects the nodes $u_i$ and $i_j$. To effectively represent the data in $T$, we utilize R-GCN (*Schlichtkrull et al., 2018*) (see the ablation study section for further details). Formally, the node $i$ within the graph at layer $(l+1)$ is represented as defined in Eq. (5).

$$n_i^{(l+1)} = \sigma\left( \sum_{r \in R} \sum_{j \in N_r} \frac{1}{Z_{i,r}} W_r^{(l)} n_j^{(l)} + W^{(l)} n_i^{(l)} \right), \tag{5}$$

where $\sigma$ denotes the activation function that introduces non-linearity into the model, $N_r$ represents all neighbor nodes connected *via* relation $r$, the normalization factor $\frac{1}{Z_{i,r}}$ adjusts the contributions from different nodes to ensure a balanced integration of information, the terms $W^{(l)}$ and $W_r^{(l)}$ represent learnable parameters, and $n_j^{(l)}$ is the representation of node $j$ at layer $l$.

### Encoding of social information

To obtain the encoded representation of social information $e_s$, we aggregate the user-item interaction graph $T$ and the user-user relationship graph $H$; the embeddings for all nodes are extracted from the topmost layers of these graphs. From graph $T$, we define $X$ as the set of all node embeddings, where $M \in X$ represents the embeddings of user nodes. Similarly, let $O$ be the set of all node embeddings in $H$.

To analyze the influence of social information on user preferences, we focus on each item $i_j$ in $C$. For each item, we identify the embeddings of all associated users in $B(j)$ as $M_u^j$ for $T$, where $M_u^j \in M$, and $O_u^j$ for $H$, where $O_u^j \in O$.

Formally, the user interaction profile $P_j$ for each $i_j$ is defined as shown in Eq. (6).

$$P_j = \mathrm{MP}(M_u^j), \tag{6}$$

where MP refers to mean pooling. This operation ensures equal contribution from all users to capture the overall characteristics the involved embeddings.

Let the user social profile $S_j$ for each $i_j$ be defined as given by Eq. (7).

$$S_j = \mathrm{MP}(O_u^j) \tag{7}$$

To obtain the enhanced user representation $E_j$ for each $i_j$ we apply Eq. (8) to stack $P_j$ and $S_j$.

$$E_j = P_j \oplus S_j \tag{8}$$

For all items $I_c$ in $C$, to obtain the set E we aggregate the enhanced user representations as expressed in Eq. (9).

$$E = \{E_j \mid i_j \in I_c\}, \tag{9}$$

Subsequently, we perform SA as described in Eq. (2), on the set E to ensure that each enhanced user representation within E is accurately weighted. By adopting this technique, we generate the social information encoded representation $e_s$.

## Applying CL to the encoded representations

After we have successfully represented multiform data through encoding, the next step is to seamlessly fuse these encoded representations to enhance the alignment and integration of information from different sources. Additionally, to build a socially sensitive fusion technique, we consider the social information $e_s$ while conducting the semantic fusion. Specifically, we incorporate $e_s$ to be fused with conversation history representation $e_c$. Thus, we perform CL for each conversation as follows: $(e_c, e_s)$. In this context, positive examples involve data representations directly related to the same conversation, while negative examples represent other data representations in batch. By fusing social

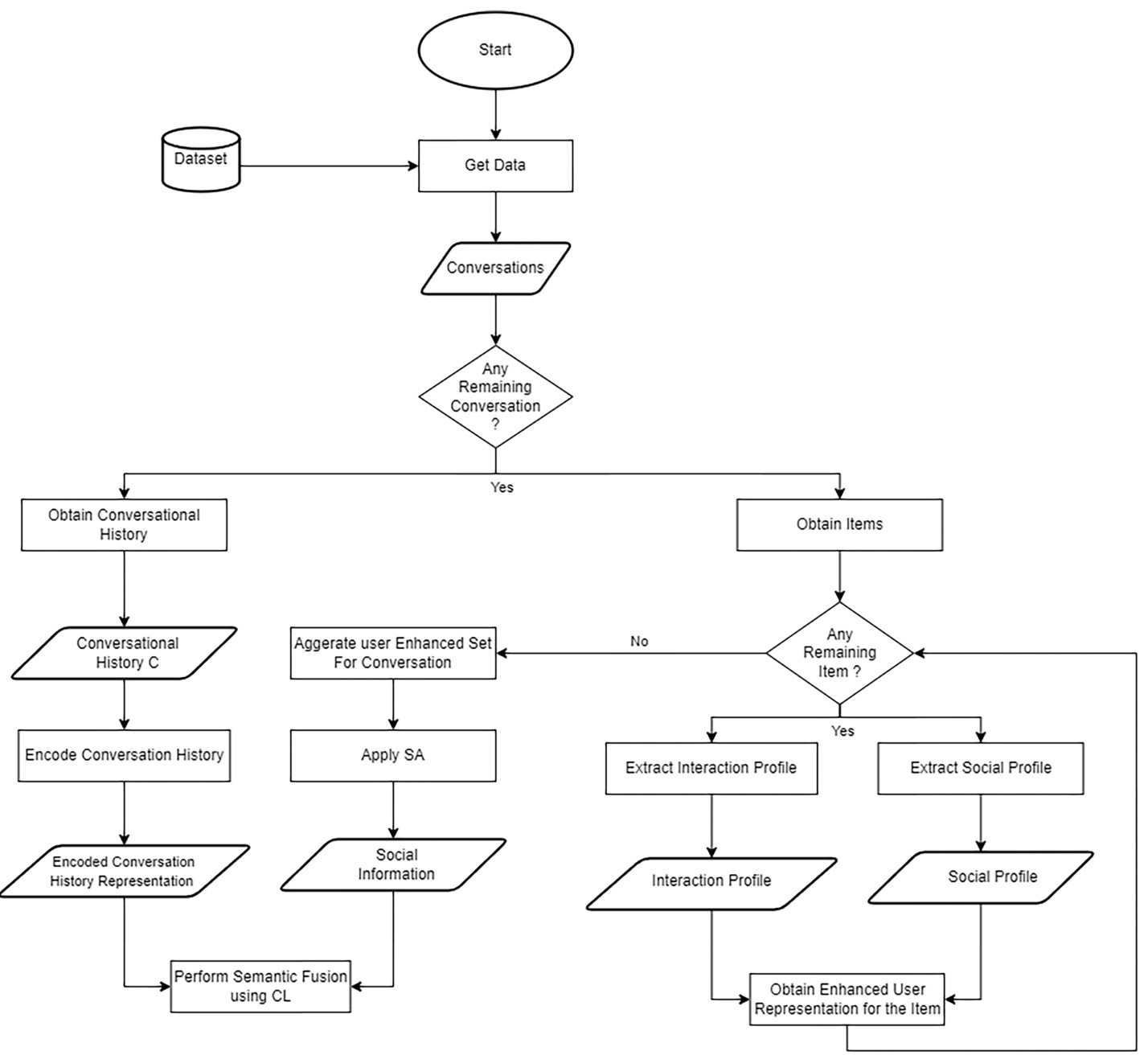

**Figure 3 Flowchart of the process of applying semantic fusion *via* CL to the encoded representations.**

information, we create a richer context, allowing the model to better understand user preferences. The contrastive loss $\mathscr{L}$ is calculated with Eq. (10). The process of applying semantic fusion using CL is illustrated in Fig. 3.

$$\mathscr{L} = \mathscr{L}_{CL}(e_c, e_s), \tag{10}$$

where $\mathscr{L}_{CL}$ is the contrastive loss function introduced by *Chen et al. (2020)* which form positive and negative samples in each batch by vertically stacking the two view embeddings

---

**Algorithm 1**   SISSF method using CL.

**Data:** user-item interaction graph $T$, user-user relationship graph $H$, and conversations $N$

**Result:** Align encoded representations for pre-trained model

1 Initialize encoder and model parameters

2 **foreach** *conversation C in N* **do**

3      Extract $P$ by accumulating the conversation history.

4      Encode $h_c$ from $P$ using Eq. (1).

5      Compute $e_c$ by applying self-attention in Eq. (2) to $h_c$.

6      Acquire $O$ by aggregating the user-user relationship graph $H$ as defined Eq. (3).

7      Acquire $X$ by aggregating the user-item interaction graph $T$ as defined Eq. (5).

8      Acquire $M$ considering all user embeddings in $X$.

9      **foreach** *item $i_j$ in $I_c$* **do**

10          Extract $M_u^j$ and $O_u^j$ from $M$ and $O$, respectively.

11          Compute user interaction profile $P_j$ by applying MP as expressed in Eq. (6) to $M_u^j$.

12          Compute user social profile $S_j$ by applying MP as expressed in Eq. (7) to $O_u^j$.

13          Compute enhanced user representation $E_j$ by stacking $P_j$ and $S_j$ as shown in Eq. (8).

14      **end foreach**

15      Acquire E by aggregating all the enhanced user representations in $C$ as expressed in Eq. (9).

16      Compute $e_s$ by applying self-attention in Eq. (2) to E.

17      Compute $\mathscr{L}$ as shown in Eq. (10).

18      Backpropagate $\mathscr{L}$.

19 **end foreach**

---

$e_c$, $e_s$ for all the batch into a single feature matrix that its size twice the batch size because we have two views. This combined matrix is then multiplied by its transpose to yield a symmetric similarity matrix, where each entry represents the cosine similarity between a pair of embeddings. By discarding the diagonal—thereby removing comparisons of an embedding with itself—we obtain pairwise similarity scores solely among different view embeddings. In this structure, those pairs where the embeddings from the two views correspond to the same conversation instance are positive examples, while all other unrelated pairs within the batches are negative examples, $\mathscr{L}_{CL}$ is defined as shown in Eqs. (11) and (12).

$$\mathscr{L}_{CL}(e_c, e_s) = \log \frac{\exp\left(\frac{\text{sim}(e_c, e_s)}{\tau}\right)}{\sum_{e_{s,i}^- \in N(e_s)} \exp\left(\frac{\text{sim}(e_c, e_{s,i}^-)}{\tau}\right)} \tag{11}$$

$$\text{sim}(e_c, e_s) = \frac{e_c^\top e_s}{||e_c|| \cdot ||e_s||} \tag{12}$$

where $N(e_s)$ represents the set of negative social information embeddings, $\tau$ represents a temperature hyperparameter, and $\text{sim}(e_c, e_s)$ denotes the cosine similarity between the two representations. Algorithm 1 presents the pseudocode for applying CL to perform semantic fusion.

## Fine tuning the rich representation to CRS's tasks

After applying CL, we obtained comprehensive representations of all components involved. These representations, stored as the weights of the pre-trained model created during semantic fusion, are fine-tuned to enhance recommendation and conversation tasks in the CRS. The process begins with selecting the relevant components for each task, followed by integrating them effectively. The following sections detail the fine-tuning process for these tasks.

### *Fine tuning for recommendation task*

To enhance the recommendation, we leverage the pre-trained model's user-item interaction graph $T$, user-user relationship graph $H$, and the conversation history encoder. Additionally, to ensure that the recommendations are highly relevant to the users' interests involved in the conversation, we have considered the representation of the users, and the conversation history up to the conversation turn. This approach allows us to achieve a more personalized recommendation and ensure social contextual awareness. Let $Y$ include all the item embeddings for $T$, where $Y \in X$. Let $Y_R$ and $Y_I$ include the items embeddings interacted with by the recommender and the seeker users, respectively, where $Y_R, Y_I \in Y$. Let $u_R$ and $u_I$ be the embeddings of the recommender and seeker users from $H$, respectively. The representation of the recommender user $e_{\text{recommender}}$, is formally defined as shown in Eq. (13).

$$e_{\text{recommender}} = \text{MP}(\text{MP}(Y_R) \oplus u_R), \tag{13}$$

Similarly, the seeker user representation $e_{\text{seeker}}$ defined as shown in Eq. (14)

$$e_{\text{seeker}} = \text{MP}(\text{MP}(Y_I) \oplus u_I) \tag{14}$$

The $e_{\text{recommender}}$ is fed into an encoder with Multi-Head Attention (MHA) cross-attention (CA) sub-layers. These sub-layers incorporate $e_{\text{seeker}}$ and $e_c$ to the encoded representation through CA as described in *Hammad, Moretti & Nojiri (2024)*. A similarly structured decoder is used, with an additional CA sub-layer to integrate the original $e_{\text{recommender}}$ and other data representations into each decoder layer during the decoding. Following *Zhou et al. (2022)*, MHA is formally defined as shown in Eq. (15).

$$\text{MHA}(Q, K, V) = \text{Concat}(\text{head}_1, \ldots, \text{head}_h)\mathbf{W}_O, \tag{15}$$

where each head is defined as expressed in Eq. (16).

$$\text{head}_i = \text{Attention}(\mathbf{QW}_i^Q, \mathbf{KW}_i^K, \mathbf{VW}_i^V), \tag{16}$$

where $\mathbf{Q}$, $\mathbf{K}$ and $\mathbf{V}$ are the query, key and value respectively, and $\mathbf{W}_i^Q$, $\mathbf{W}_i^K$, and $\mathbf{W}_i^V$ are the learnable weights.

The $e_{recommender}$ is processed by the encoder, as shown in Eqs. (17)–(21):

$$A_0^n = \text{MHA}(E^{n-1}, E^{n-1}, E^{n-1}), \tag{17}$$
$$A_1^n = \text{MHA}(A_0^n, e_{seeker}, e_{seeker}), \tag{18}$$
$$A_2^n = \text{MHA}(A_1^n, e_c, e_c), \tag{19}$$
$$E^n = \text{FFN}(A_2^n), \tag{20}$$
$$\text{FFN}(x) = \max(0, xW_1 + b_1)W_2 + b_2, \tag{21}$$

where $E^{n-1}$ is the encoded representation of $e_{recommender}$ at the $(n-1)$-th layer, $A_0^n$ is the representation after applying SA with $E^{n-1}$, $A_1^n$ is the representation after applying CA with $e_{seeker}$, $A_2^n$ is the representation after applying CA with FFN is a feedforward neural network for dimension mapping, and $E^n$ is the output of the encoder at the topmost layer.

Subsequently, the output of the encoder is processed by the decoder as shown in Eqs. (22)–(26):

$$B_0^n = \text{MHA}(R^{n-1}, R^{n-1}, R^{n-1}), \tag{22}$$
$$B_1^n = \text{MHA}(B_0^n, e_{recommender}, e_{recommender}), \tag{23}$$
$$B_2^n = \text{MHA}(B_1^n, e_{seeker}, e_{seeker}), \tag{24}$$
$$B_3^n = \text{MHA}(B_2^n, e_c, e_c), \tag{25}$$
$$R^n = \text{FFN}(B_3^n), \tag{26}$$

where $R^{n-1}$ is the decoded representation of the encoder output $E^n$ at the previous layer $n-1$, $B_0^n$ is the representation after applying SA with $R^{n-1}$, $B_1^n$ is the representation after applying CA with $e_{recommender}$, $B_2^n$ is the representation after applying CA with $e_{seeker}$, $B_3^n$ is the representation after applying CA with $e_c$, and $R^n$ is the output of the decoder at the final layer $n$, which represents the probabilities of the recommended items after applying softmax.

Let $P_{rec}(i)$ be the probability distribution for the recommended item at the index $i$ as defined in Eqs. (27) and (28).

$$P_{rec}(i) = \text{softmax}(R^n)[i], \tag{27}$$
$$\sum P_{rec}(i) = 1, \tag{28}$$

Lastly, to fine-tune the rich representations for the recommendation task, following *Zhou et al. (2020a, 2022)*, we apply cross-entropy loss to the probabilities of the recommended items. In our case, we consider one recommended item per conversation as given by Eq. (29).

$$\mathscr{L}rec = -\sum_{n=1}^{N} \sum_{i=1}^{IR} y_{ni} \cdot \log(P_{rec}^n(i)), \tag{29}$$

where $N$ is the number of conversations, $IR$ represents the items being recommended, and $y_{ni}$ is the ground truth.

### Fine tuning for conversational task

For the conversation task, all models within the pre-trained framework are utilized. The approach aligns with the methodology detailed in the previous section, with adjustments

inspired by *Zhou et al. (2020a, 2022)* to fine-tune the pre-trained model for response generation. Specifically, an encoder, as described in Eq. (1) of a standard transformer (*Vaswani et al., 2017*), is used to encode the responses in training and inference. Additionally, in the decoder, different additional MHA sub-layers with CA sub-layers are used to incorporate the social information $e_s$, conversation history representation $e_c$, and the contextual embeddings $h_c$ from the pre-trained model during the decoding process as defined in Eqs. (30)–(34).

$$C_0^n = \text{MHA}(R^{n-1}, R^{n-1}, R^{n-1}), \tag{30}$$

$$C_1^n = \text{MHA}(C_0^n, e_s, e_s), \tag{31}$$

$$C_2^n = \text{MHA}(C_1^n, e_c, e_c), \tag{32}$$

$$C_3^n = \text{MHA}(C_2^n, h_c, h_c), \tag{33}$$

$$R^n = \text{FFN}(C_3^n), \tag{34}$$

where $C_0^n$ is the representation after applying self-attention with the decoder output at the previous layer $R^{n-1}$, $C_1^n$ is the representation after applying cross-attention with $e_s$, $C_2^n$ is the representation after applying cross-attention with $e_c$, $C_3^n$ is the representation after applying cross-attention with $h_c$, and $R^n$ is the representation of the decoder output at the $n$-th layer.

The copying mechanism used in both *Zhou et al. (2020a, 2022)* is also adopted to copy useful information from user information through integrating enhanced user representation set E while generating responses to enhance performance and personalization. Formally, the probability of generating token $w_j$ given tokens $w_1, \ldots, w_{j-1}$ is defined in Eq. (35).

$$P(w_j|w_1, \ldots, w_{j-1}) = P_{gen}(w_j|R^n) + P_{copy}(w_j|R^n, \text{E}), \tag{35}$$

where $P_{gen}$ is the generation probability over all the vocabulary given the decoder output at the final layer $R^n$, and $P_{copy}$ is the copy probability given $R^n$ and E.

The loss function used for fine-tuning is the same as the one used in *Zhou et al. (2022)*, where a weighted cross-entropy loss is used to improve the response generation as defined in Eqs. (36) and (37).

$$\mathscr{L}\text{gen} = -\frac{1}{m}\sum_{j=1}^{m} \log(\alpha_{w_j} P(w_j|w_1, \ldots, w_{j-1})), \tag{36}$$

$$\alpha_{wj} = \begin{cases} \max\left(\gamma, \frac{\beta}{f_{w_j}}\right), & \text{if } f_{wj} \geq \beta \\ 1, & \text{otherwise} \end{cases}, \tag{37}$$

where $m$ is the total number of tokens in the generated response, $\alpha_{wj}$ is the weight of the token depending on its frequency, $f_{w_j} \in \{1, 2, 3, \ldots\}$ is the word frequency, $\gamma$ is a random number in the range $[0, 1]$ that balances the impact of word frequency during response generation, and $\beta \in \{1, 2, 3, \ldots\}$ is the threshold of the word frequency.

Furthermore, we have used two decoding techniques. Greedy Search (*Radford et al., 2019*) is adopted to generate $w_j$ during training as defined in Eq. (38), while in the inference

stage, we use Nucleus Sampling (*Holtzman et al., 2020*) to select $w_j$ as defined in Eqs. (39)–(42).

$$w_j = \underset{w_j}{\operatorname{argmax}} \; \alpha_{w_j} P(w_j | w_1, \ldots, w_{j-1})), \tag{38}$$

where $\operatorname{argmax}_{w_j}$ is the argument of the maximum function.

$$w_j \sim P'(w_j \mid w_1, \ldots, w_{j-1}), \tag{39}$$

$$P'(w_j \mid w_1, \ldots, w_{j-1}) = \begin{cases} \frac{\alpha_{w_j} P(w_j \mid w_1, \ldots, w_{j-1})}{p'} & \text{if } w_j \in V(p) \\ 0 & \text{otherwise} \end{cases}, \tag{40}$$

$$p' = \sum_{w_j \in V(p)} \alpha_{w_j} P(w_j \mid w_1, \ldots, w_{j-1}), \tag{41}$$

$$V(p) = \left\{ w_j \mid \sum_{w_j \in V} \alpha_{w_j} P(w_j \mid w_1, \ldots, w_{j-1}) \geq p \right\}, \tag{42}$$

where $p$ is a random number in the range $[0, 1]$ that represents the cumulative probability threshold, $V$ is the total vocabulary, $V(p)$ is the smallest set of tokens in which their cumulative probability is greater than or equal to $p$, $p'$ is a rescaling factor that is the total probability of all tokens in $V(p)$ used to normalize the selected tokens' probabilities, and $P'(w_j \mid w_1, \ldots, w_{j-1})$ is the rescaled probability distribution used to sample $w_j$.

# EXPERIMENT

In this section, we discuss the experiment in detail, starting with the experiment setup, followed by the evaluation of the experiment on all tasks on all datasets. Next, we perform a statistical analysis to check the validity of our findings. Then, we conduct hyperparameter analysis on the ReDial dataset (*Li et al., 2018*) to find the optimal configuration and conclude with an ablation study to prove the effectiveness of the proposed framework on the ReDial dataset (*Li et al., 2018*).

## Experiment setup

This section presents the experiment setup, including the datasets, annotation process, defining baselines, choosing evaluation metrics, and stating the implementation details.

### Datasets

The ReDial dataset (*Li et al., 2018*) and the INSPIRED dataset (*Hayati et al., 2020*). The ReDial dataset (*Li et al., 2018*) was created by Amazon Mechanical Turk (AMT) and serves as a conversational dataset for movie recommendations. It consists of 10,000 conversations with 182,150 utterances associated with 51,699 movies in the English language. The INSPIRED dataset (*Hayati et al., 2020*) is much smaller, consisting of 1,001 conversations with 35,811 utterances associated with 1,783 items. Both datasets are English CRS datasets. Each conversation involves a dialogue between two users, where one user acts as the recommender and the other acts as the seeker. The users might fluctuate their roles across conversations within the dataset, so mutual recommendations of movies take place.

**Table 1 Statistics of users and their roles for all datasets.**

| Dataset | Seeker | Recommender | Both | Total |
|---|---|---|---|---|
| ReDial | 764 | 856 | 508 | 1,112 |
| INSPIRED | 777 | 721 | 109 | 1,389 |

Table 1 shows the statistics of users for all datasets. Following the baselines, no sampling techniques are used for balancing user roles.

Our experiments require a user-item interaction graph and a user-user relationship graph for the datasets. However, the ReDial (*Li et al., 2018*) and the INSPIRED (*Hayati et al., 2020*) datasets only include conversational data. We leverage the available metadata in the ReDial dataset (*Li et al., 2018*) to infer reasonable interaction information. However, the INSPIRED dataset (*Hayati et al., 2020*) does not contain this metadata about each user's mentioned movies and the details of each movie interaction, whether liked, suggested, or watched. To address this, we manually annotate each movie mentioned in conversations to add users' movie interaction information to the INSPIRED dataset (*Hayati et al., 2020*). The metadata from both datasets are then used to generate ratings for both datasets separately. The process of annotating the dataset is described in detail in the next subsection.

We initiate the process by extracting an interaction matrix that records positive and negative interactions, specified by the liking or disliking of items mentioned by each user in the conversations. We form a social relationships matrix conditioned upon the criterion that conversing users demonstrate at least one shared interest, according to *Hayati et al. (2020)*. In addition, as the recommender user influences the seekers who converse with them, we also establish friend relationships among the seekers who interacted with the same recommender under the same condition mentioned earlier. Therefore, we ensure that the connections formed are based on engagement and shared interests. Lastly, we utilize NGCF (*Wang et al., 2019*) to generate ratings for all items. The NGCF relies on the interaction information to predict ratings. These ratings contribute to creating the user-item interaction graph $T$.

We construct the social relationship graph $H$ using GTO (*Shi et al., 2021*), where nodes represent users in the social relationships matrix. Similarly, we construct $T$ using R-GCN (*Schlichtkrull et al., 2018*) by replacing the positive and negative interactions with the inferred ratings we obtained in the previous step, where nodes represent users and items, with directed edges from user nodes to item nodes.

### Annotation process

Each dialogue in the ReDial dataset (*Li et al., 2018*) contains additional fields, such as a dictionary to map movie IDs to their names and dictionaries that associate movie IDs with labels indicating whether a movie is seen, liked, or suggested by the initiator or respondent. These labels include suggested (distinguishing who suggested the movie), seen (indicating whether the movie was watched), and liked (denoting if the user liked the movie). These labels are essential for the construction of the required graphs for SISSF. As the INSPIRED

**conv_id: 20191129-120149_55_live.pkl**

**RECOMMENDER:** Hello

**SEEKER:** hey

**RECOMMENDER:** how are you doing this lovely afternoon

**SEEKER:** pretty good thanks and you ? So I'm looking for a movie with lot of twist. what do you recommend ?

**RECOMMENDER:** Okay, what do you think about the movie @7701

**SEEKER:** is it a romance movie ?

**RECOMMENDER:** Yes it is, you would like it if you are into romance

**SEEKER:** I am more a type than romance

**RECOMMENDER:** - would that be leaning toward horror

**SEEKER:** could be, thriller also, like heist or crime

**RECOMMENDER:** I'm sure you're going to love @436 I did see the movie thrice.

**SEEKER:** I'm watching the trailer. Looks pretty good. the music is great.

**RECOMMENDER:** Awesome. I'm glad you like it

**SEEKER:** Good recommendation thanks ;)

**RECOMMENDER:** you are welcome

## Entry Form

**Figure 4** Each sub-form allows the annotators to assign movies to the corresponding user by selecting the specific movie from a drop-down menu. Once a movie is selected, it is labeled according to the context of the conversation, with seen, liked, and suggested labels for each user.

dataset (*Hayati et al., 2020*) initially does not contain these labels, we manually conduct the annotation process. Figure 4 shows the user interface we created, which is designed to display conversations with highlighted movie IDs coupled with a dynamic form that

contains two dynamic sub-forms, where the annotators can add movies for the corresponding users.

### Baselines

We evaluate our CRS based on recommendation and conversational tasks, using benchmark CRSs and stand-alone models as baselines.

- **Popularity:** The items in the *corpus* are ranked based on the frequency of recommendations they receive in the training set. This ranking is based on popularity.
- **TextCNN** (*Kim, 2014*): CCN extracts user preferences from text to recommend items.
- **Transformer** (*Vaswani et al., 2017*): Generates conversational responses using a transformer-based encoder-decoder model.
- **KBRD** (*Chen et al., 2019*): It uses the seq2seq model based on the transformer (*Vaswani et al., 2017*) and a general KG from DBpedia to improve the recommendation integration and the response generation components.
- **KGSF** (*Zhou et al., 2020a*): Enhances the CRS by adapting a way to enhance the semantic fusion between the external KGs and conversation history through mutual information maximization.
- **ReDial** (*Li et al., 2018*): Uses the seq2seq model based on HRED (*Serban et al., 2016*) for conversation generalization with a switch decoder (*Gülçehre et al., 2016*) to decide when to recommend or generate a token. The recommender is an autoencoder (*Sedhain et al., 2015*), and a sentiment analysis unit is used to check the user's feedback. The model is proposed with the ReDial dataset (*Li et al., 2018*).
- **INSPIRED** (*Hayati et al., 2020*): Proposes the INSPIRED dataset (*Hayati et al., 2020*) for CRS encoded with sociable strategies to create a CRS that leverages social relationships.
- **INSPIRED2** (*Ahtsham & Dietmar, 2022*): Proposes the INSPIRED2 dataset (*Ahtsham & Dietmar, 2022*) for CRS, which is an improved version of the INSPIRED dataset (*Hayati et al., 2020*).
- **UniCRS** (*Wang et al., 2022*): Utilize PL to guide PLMs for recommendation and conversation generations.
- **C2-CRS** (*Zhou et al., 2022*): Improves the semantic fusion by prompting a novel way to fuse many external data with the conversation history through CL (*Jing et al., 2023*).
- **VRICR** (*Zhang et al., 2023*): Uses variational Bayesian technique to infer the missing links among entities related to the subgraphs that are specific to a given conversation.

Popularity and TextCNN (*Kim, 2014*) serve as benchmarks exclusively for the recommendation task, while the Transformer (*Vaswani et al., 2017*) is only evaluated for the conversational task. The CRS models KBRD (*Chen et al., 2019*), KGSF (*Zhou et al., 2020a*), C2-CRS (*Zhou et al., 2022*), VRICR (*Zhang et al., 2023*), INSPIRED (*Hayati et al., 2020*), INSPIRED2 (*Ahtsham & Dietmar, 2022*), UniCRS (*Wang et al., 2022*), and ReDial (*Li et al., 2018*) are used to evaluate both recommendation and conversational tasks.

Since not all baselines are tested on every dataset, we grouped them by dataset for the SISSF evaluation. On the ReDial dataset (*Li et al., 2018*), we consider the following

**Table 2 Overview of CRS models and their key features.**

| Model | Year | Core technology | Main innovation | Reference |
|---|---|---|---|---|
| ReDial | 2018 | Seq2seq based on HEAD, Autoencoder | Switch mechanism between generation and recommendation | Li et al. (2018) |
| KBRD | 2019 | Seq2seq based on Transformer, KG | Knowledge-enhanced dialogue policy | Chen et al. (2019) |
| KGSF | 2020 | Seq2seq, KG, MIM | Semantic fusion with external KG | Zhou et al. (2020a) |
| INSPIRED | 2020 | Social Strategies | Sociable recommendation dialogue | Hayati et al. (2020) |
| INSPIRED2 | 2022 | Social Strategies, Improved Entity Annotation | Sociable recommendation dialogue with improvement of entity annotation | Ahtsham & Dietmar (2022) |
| UniCRS | 2022 | PL, PLMs | Leverages knowledge-enhanced prompt learning to combine the recommendation and conversation | Wang et al. (2022) |
| C2-CRS | 2022 | Seq2seq, KG, CL | Coarse-to-fine semantic fusion | Zhou et al. (2022) |
| VRICR | 2023 | Seq2seq, A Bayesian method, KG | Adapting reasoning process based on the dialogue Context | Zhang et al. (2023) |
| SISSF (Ours) | 2024 | Seq2seq, Social information, CL | Social information sensitive fusion | Current work |

baselines: Popularity, TextCNN (*Kim, 2014*), Transformer (*Vaswani et al., 2017*), KBRD (*Chen et al., 2019*), KGSF (*Zhou et al., 2020a*), C2-CRS (*Zhou et al., 2022*), VRICR (*Zhang et al., 2023*), UniCRS (*Wang et al., 2022*), and ReDial (*Li et al., 2018*), while on the INSPIRED dataset (*Hayati et al., 2020*), the baselines include KBRD (*Chen et al., 2019*), KGSF (*Zhou et al., 2020a*), INSPIRED (*Hayati et al., 2020*), INSPIRED2 (*Ahtsham & Dietmar, 2022*), UniCRS (*Wang et al., 2022*), and ReDial (*Li et al., 2018*). Table 2 summarizes the details of various CRS models.

### Evaluation metrics

We adopt different evaluation metrics tailored to the distinct nature of each task. For the recommendation task, according to the baseline methods chosen for this work, most of them primarily focus on Recall as the evaluation metric. To ensure consistency with these baselines, we opted to use Recall@K ($k = 1, 10, 50$). Similarly, for the conversational task, we apply human evaluation for the ReDial dataset (*Li et al., 2018*) and automatic evaluation for both datasets (*Hayati et al., 2020*; *Li et al., 2018*). During the automatic evaluation, we measure the diversity of generated utterances using Distinct $n$-gram ($n = 2, 3, 4$), while during the human evaluation, we used three annotators on a scale from 0 to 2 to score the generated responses in two aspects, namely *Fluency* and *Informativeness*. The average score is computed for all the annotators to get the final performance score.

### Implementation details

We follow Algorithm 1 with PyTorch (https://pytorch.org) and WSDM2022-C2CRS (*Zhou et al., 2022*) (https://github.com/RUCAIBox/WSDM2022-C2CRS) to implement our approach. The embedding dimensionalities are set to 300 for token embeddings and 768 for the user dimensions. For the user-item interaction graph $T$ and user-user relationship graph $H$, we set the layer number and the normalization constant to 1. We use the Adam optimizer (*Kingma & Ba, 2015*) with beta values of (0.9, 0.999), and batch size of 300. The learning rates for our model are carefully chosen for each model to ensure

**Table 3 The recommendation task results on the ReDial dataset.** Results marked with an asterisk (*) are statistically significant according to the Wilcoxon rank sum test ($p$-value $< 0.05$).

| Dataset | ReDial | | |
|---|---|---|---|
| **Model** | **R@1** | **R@10** | **R@50** |
| Popularity | 0.012 | 0.061 | 0.179 |
| TextCNN | 0.013 | 0.068 | 0.191 |
| ReDial | 0.024 | 0.140 | 0.320 |
| KBRD | 0.031 | 0.150 | 0.336 |
| KGSF | 0.039 | 0.183 | 0.378 |
| UniCRS | 0.051 | 0.224 | 0.428 |
| C2-CRS | 0.053 | 0.233 | 0.407 |
| VRICR | 0.057 | 0.251 | 0.416 |
| SISSF | 0.062* | 0.245 | 0.437* |

**Table 4 The recommendation task results on the INSPIRED dataset.** Results marked with an asterisk (*) are statistically significant according to the Wilcoxon rank sum test ($p$-value $< 0.05$).

| Dataset | INSPIRED | | |
|---|---|---|---|
| **Model** | **R@1** | **R@10** | **R@50** |
| KGSF | 0.002 | 0.021 | 0.074 |
| INSPIRED | 0.015 | 0.055 | 0.153 |
| INSPIRED2 | 0.019 | 0.073 | 0.185 |
| UniCRS | 0.029 | 0.095 | 0.240 |
| SISSF | 0.046* | 0.129* | 0.269* |

optimal performance. During the semantic fusion and fine-tuning of the recommendation task, we use a learning rate of 0.00001. For the conversational task, we start with 0.0001 and then resume learning with 0.00001 for the ReDial dataset (*Li et al., 2018*) and maintain a learning rate of 0.0001 for the INSPIRED dataset (*Hayati et al., 2020*). Regarding the CL, we set a temperature of 0.09 during semantic fusion and 0.13 during fine-tuning. For the transformer model, we set the number of layers and heads to 4 for both transformers used by both tasks. For the calculation of $\mathscr{L}$gen, we set $\gamma$ to 0.3 and $\beta$ to 100. We set the cumulative probability threshold $p$ to 0.95 during the inference process. Following the section Applying CL to the Encoded Representations, the stage of aligning the data representation to minimize the loss is conducted during CL. In addition, during the fine-tuning stage, the section Fine Tuning the Rich Representation to CRS's Tasks is considered to optimize the loss for each specific task. The project code and data are available at https://doi.org/10.5281/zenodo.15368530.

## Evaluation on recommendation task

This section demonstrates the effectiveness of the proposed framework on the recommendation task. We conducted a series of experiments to measure the performance on both datasets. Table 3 presents the performance metrics against different benchmarks

for the ReDial dataset (*Li et al., 2018*), while Table 4 presents the performance metrics against different benchmarks related to the INSPIRED dataset (*Hayati et al., 2020*).

For the ReDial dataset (*Li et al., 2018*), overall, CRSs demonstrate superior performance compared to traditional recommender systems because CRSs combine both the recommender component and the conversation component to update user preferences dynamically based on the conversation history. In contrast, traditional recommendation models lack the adaptability to capture user preferences in real time. TextCNN (*Kim, 2014*) performs slightly better than Popularity as it recommends items based on contextual information, whereas Popularity lacks personalization as it treats all users the same, relying solely on item popularity. KBRD (*Chen et al., 2019*), KGSF (*Zhou et al., 2020a*), UniCRS (*Wang et al., 2022*), C2-CRS (*Zhou et al., 2022*), and VRICR (*Zhang et al., 2023*) outperform ReDial (*Li et al., 2018*) by integrating external datasets using different techniques. VRICR (*Zhang et al., 2023*) demonstrates the highest performance due to its effective integration of external datasets, followed by C2-CRS (*Zhou et al., 2022*), UniCRS (*Wang et al., 2022*), KGSF (*Zhou et al., 2020a*), and then KBRD (*Chen et al., 2019*), each showing a gradual performance improvement based on the nature of the external datasets and the techniques. Our model shows a performance increase compared to all benchmarks, particularly excelling on R@1 and R@50 compared to VRICR (*Zhang et al., 2023*).

For the INSPIRED dataset (*Hayati et al., 2020*), INSPIRED (*Hayati et al., 2020*) shows better performance compared to KGSF (*Zhou et al., 2020a*) by considering the social strategies. In contrast, INSPIRED2 (*Ahtsham & Dietmar, 2022*) outperforms the INSPIRED (*Hayati et al., 2020*) as it improves the quality of entity annotation. While UniCRS (*Wang et al., 2022*) achieves the highest performance among the baselines, SISSF outperforms all models across all metrics. This is achieved by integrating the social relationship inferred by combining the interactions and user relationships with the conversation history, which improves the model's capability to deduce user preferences, leading to accurate recommendations.

## Evaluation on conversational task

In this section, we acknowledge the improvement of the proposed framework in the automatic and human evaluation of the conversational task compared to the best baselines.

### Automatic evaluation

Tables 5 and 6 present the results of the automatic evaluation of the conversational task on the ReDial (*Li et al., 2018*) and the INSPIRED (*Hayati et al., 2020*) datasets, respectively. For the ReDial (*Li et al., 2018*) dataset, the Transformer (*Vaswani et al., 2017*) performs poorly due to its slow dependency on the conversation history to generate results. ReDial (*Li et al., 2018*) performs better than the Transformer (*Vaswani et al., 2017*), which benefits from integrating the recommender component during response generation with the help of a switch mechanism. KBRD (*Chen et al., 2019*) outperforms ReDial (*Li et al., 2018*) on ReDial (*Li et al., 2018*) and INSPIRED (*Hayati et al., 2020*) datasets as it utilizes the KG to

**Table 5 Results of the Automatic evaluation on the conversational task on the ReDial dataset.** Results marked with asterisk (*) are statistically significant according to the Wilcoxon rank sum test (*p*-value < 0.05).

| Dataset | ReDial | | |
|---|---|---|---|
| Model | Distinct-2 | Distinct-3 | Distinct-4 |
| Transformer | 0.067 | 0.139 | 0.227 |
| ReDial | 0.082 | 0.143 | 0.245 |
| KBRD | 0.086 | 0.153 | 0.265 |
| KGSF | 0.114 | 0.204 | 0.282 |
| UniCRS | 0.142 | 0.255 | 0.302 |
| C2-CRS | 0.163 | 0.291 | 0.417 |
| VRICR | 0.165 | 0.292 | 0.482 |
| SISSF | 4.223* | 5.595* | 6.155* |

**Table 6 Results of Automatic evaluation on the conversational task on the INSPIRED dataset.** Results marked with asterisk (*) are statistically significant according to the Wilcoxon rank sum test (*p*-value < 0.05).

| Dataset | INSPIRED | | |
|---|---|---|---|
| Model | Distinct-2 | Distinct-3 | Distinct-4 |
| ReDial | 0.153 | 0.255 | 0.397 |
| KBRD | 0.223 | 0.415 | 0.616 |
| UniCRS | 1.424 | 2.790 | 3.628 |
| SISSF | 2.061* | 4.293* | 6.242* |

enrich the contextual information. KGSF (*Zhou et al., 2020a*) performs better than KBRD (*Chen et al., 2019*) due to semantic fusion to bridge the semantic gap between the KGs and conversational history, leading to richer contextual representation. UniCRS (*Wang et al., 2022*) surpasses KGSF (*Zhou et al., 2020a*) by incorporating fused information to prompt the PLMs. *Zhou et al. (2020a)* C2-CRS (*Zhou et al., 2022*) outperforms UniCRS (*Wang et al., 2022*) by adopting a new method of semantic fusion and involving more external data. VRICR (*Zhang et al., 2023*) achieves the highest results on all metrics among the baselines due to the more effective utilization of external data. Furthermore, our model outperforms all the baselines on all metrics on all datasets, primarily due to implementing an intuitive semantic fusion technique that integrates social information, providing additional context about user preferences and interests for more relevant responses. Additionally, utilizing the top-p sampling (*Holtzman et al., 2020*) technique enhances the diversity of response generation.

### Human evaluation

The human evaluation results on conversational task on the ReDial dataset (*Li et al., 2018*) are presented in Table 7. First, VRICR (*Zhang et al., 2023*) and C2-CRS (*Zhou et al., 2022*)

**Table 7 Results of human evaluation on the conversational task on the ReDial dataset.** Results marked with asterisk (*) are statistically significant according to the Wilcoxon rank sum test ($p$-value $< 0.05$).

| Dataset | ReDial | |
|---|---|---|
| **Model** | **Fluency** | **Informativeness** |
| Transformer | 0.97 | 0.92 |
| ReDial | 1.35 | 1.04 |
| KBRD | 1.28 | 1.15 |
| KGSF | 1.48 | 1.37 |
| UniCRS | 1.50 | 1.41 |
| C2-CRS | 1.52 | 1.47 |
| VRICR | 1.57 | 1.53 |
| SISSF | 1.81* | 1.63* |

achieve the best results compared to all baselines. This is followed by UniCRS (*Wang et al., 2022*) that utilizes PL to integrate the fused KG with PLMs. Next, KGSF (*Zhou et al., 2020a*), where a KG-enhanced Transformer decoder injects the data from the rich representations generated by fusing conversational text and items *via* a knowledge graph. KBRD (*Chen et al., 2019*) achieved slightly better metrics results than ReDial (*Li et al., 2018*) by promoting low-frequency tokens by leveraging the knowledge graph. In contrast, ReDial (*Li et al., 2018*) used a pre-trained encoder, which exceeded the results of the Transformer (*Vaswani et al., 2017*). Finally, SISSF outperforms all the benchmarks on all metrics by leveraging CL to bridge the gap between conversational history and social information. Additionally, a Transformer decoder integrates the generated compact representation with a weighting mechanism to improve the informativeness of responses. This is enhanced by the top-p sampling technique (*Holtzman et al., 2020*), which has been proven to generate fluent, human-like responses, reflecting the significant improvement of our model's fluency.

## Statistical analysis and comparison

To demonstrate the efficiency of the proposed model, statistical analysis was conducted on the recommendation tasks using the ReDial (*Li et al., 2018*) and INSPIRED (*Hayati et al., 2020*) datasets. The population size is 30 with 500 iterations and 10 independent runs. Tables 8 and 9 report statistical summaries of model performance on the ReDial (*Li et al., 2018*) and INSPIRED (*Hayati et al., 2020*) datasets, respectively. The performance of SSISF was compared to other baselines using Wilcoxon rank sum test. The $p$-values are recorded in Tables 10 and 11 for the ReDial (*Li et al., 2018*) and INSPIRED (*Hayati et al., 2020*) datasets, respectively. Table 10 shows $p$-values are less than 0.05 for R@1 and R@50, while Table 11 shows all $p$-values are less than 0.05 for all metrics. This clearly shows that SISSF outperforms other baselines in the recommendation task. In cases where no significant difference was found, "N/A" is recorded.

**Table 8  Statistical summary of evaluation results on the ReDial dataset.**

| Dataset | ReDial | | | |
|---|---|---|---|---|
| **Model** | **Stat** | **R@1** | **R@10** | **R@50** |
| Popularity | Mean | 9.95E−3 | 6.17E−2 | 1.74E−1 |
| | Median | 9.47E−3 | 6.26E−2 | 1.67E−1 |
| | STD | 4.27E−3 | 1.23E−2 | 3.02E−2 |
| TextCNN | Mean | 1.08E−2 | 6.87E−2 | 1.86E−1 |
| | Median | 1.03E−2 | 6.97E−2 | 1.79E−1 |
| | STD | 4.63E−3 | 1.38E−2 | 3.22E−2 |
| ReDial | Mean | 1.99E−2 | 1.42E−1 | 3.12E−1 |
| | Median | 1.90E−2 | 1.44E−1 | 2.99E−1 |
| | STD | 8.54E−3 | 2.83E−2 | 5.39E−2 |
| KBRD | Mean | 2.57E−2 | 1.52E−1 | 3.27E−1 |
| | Median | 2.45E−2 | 1.54E−1 | 3.14E−1 |
| | STD | 1.10E−2 | 3.03E−2 | 5.66E−2 |
| KGSF | Mean | 2.90E−2 | 1.85E−1 | 3.52E−1 |
| | Median | 2.76E−2 | 1.88E−1 | 3.38E−1 |
| | STD | 1.25E−2 | 3.70E−2 | 6.08E−2 |
| UniCRS | Mean | 3.04E−2 | 1.89E−1 | 3.69E−1 |
| | Median | 2.90E−2 | 1.91E−1 | 3.54E−1 |
| | STD | 1.31E−2 | 3.78E−2 | 6.38E−2 |
| C2-CRS | Mean | 3.16E−2 | 1.99E−1 | 3.51E−1 |
| | Median | 3.01E−2 | 2.03E−1 | 3.37E−1 |
| | STD | 1.36E−2 | 3.99E−2 | 6.07E−2 |
| VRICR | Mean | 3.17E−2 | 2.03E−1 | 3.69E−1 |
| | Median | 3.02E−2 | 2.06E−1 | 3.54E−1 |
| | STD | 1.36E−2 | 4.06E−2 | 6.37E−2 |
| SISSF | Mean | 5.14E−2 | 2.48E−1 | 4.26E−1 |
| | Median | 4.89E−2 | 2.51E−1 | 4.09E−1 |
| | STD | 2.21E−2 | 4.96E−2 | 7.36E−2 |

**Table 9  Statistical summary of evaluation results on the INSPIRED dataset.**

| Dataset | INSPIRED | | | |
|---|---|---|---|---|
| **Model** | **Stat** | **R@1** | **R@10** | **R@50** |
| KGSF | Mean | 1.49E−3 | 1.93E−2 | 6.19E−2 |
| | Median | 1.26E−3 | 1.87E−2 | 6.10E−2 |
| | STD | 9.90E−4 | 6.52E−3 | 9.90E−3 |
| INSPIRED | Mean | 1.12E−2 | 5.05E−2 | 1.28E−1 |
| | Median | 9.45E−3 | 4.90E−2 | 1.26E−1 |
| | STD | 7.39E−3 | 1.71E−2 | 2.05E−2 |
| INSPIRED2 | Mean | 1.42E−2 | 6.71E−2 | 1.55E−1 |
| | Median | 1.20E−2 | 6.50E−2 | 1.52E−1 |
| | STD | 9.36E−3 | 2.27E−2 | 2.48E−2 |

| Table 9 (continued) | | | | |
|---|---|---|---|---|
| Dataset | INSPIRED | | | |
| Model | Stat | R@1 | R@10 | R@50 |
| UniCRS | Mean | 1.55E−2 | 7.27E−2 | 1.55E−1 |
| | Median | 1.31E−2 | 7.05E−2 | 1.52E−1 |
| | STD | 1.02E−2 | 2.46E−2 | 2.47E−2 |
| SISSF | Mean | 3.43E−2 | 1.18E−1 | 2.25E−1 |
| | Median | 2.90E−2 | 1.15E−1 | 2.22E−1 |
| | STD | 2.27E−2 | 4.00E−2 | 3.60E−2 |

**Table 10 $p$-values of the Wilcoxon rank sum test with 0.05 Significance for SISSF against other baselines on the recommendation task on the ReDial dataset.**

| Dataset | ReDial | | |
|---|---|---|---|
| Comparison | R@1 | R@10 | R@50 |
| SISSF *vs.* Popularity | $1.57 \times 10^{-4}$ | $1.57 \times 10^{-4}$ | $1.57 \times 10^{-4}$ |
| SISSF *vs.* TextCNN | $1.57 \times 10^{-4}$ | $1.57 \times 10^{-4}$ | $1.57 \times 10^{-4}$ |
| SISSF *vs.* ReDial | $8.81 \times 10^{-4}$ | $3.81 \times 10^{-4}$ | $1.15 \times 10^{-3}$ |
| SISSF *vs.* KBRD | $5.16 \times 10^{-3}$ | $3.81 \times 10^{-4}$ | $5.16 \times 10^{-3}$ |
| SISSF *vs.* KGSF | $1.56 \times 10^{-2}$ | $2.24 \times 10^{-2}$ | $1.91 \times 10^{-2}$ |
| SISSF *vs.* UniCRS | $2.83 \times 10^{-2}$ | $3.15 \times 10^{-2}$ | $4.94 \times 10^{-2}$ |
| SISSF *vs.* C2-CRS | $4.13 \times 10^{-2}$ | $3.43 \times 10^{-2}$ | $1.91 \times 10^{-2}$ |
| SISSF *vs.* VRICR | $4.13 \times 10^{-2}$ | N/A | $4.94 \times 10^{-2}$ |

**Table 11 $p$-values of the Wilcoxon rank sum test with 0.05 Significance for SISSF against other baselines on the recommendation task on the INSPIRED dataset.**

| Dataset | INSPIRED | | |
|---|---|---|---|
| Comparison | R@1 | R@10 | R@50 |
| SISSF *vs.* KGSF | $1.12 \times 10^{-4}$ | $1.12 \times 10^{-4}$ | $1.12 \times 10^{-4}$ |
| SISSF *vs.* INSPIRED | $5.16 \times 10^{-3}$ | $2.85 \times 10^{-4}$ | $1.12 \times 10^{-4}$ |
| SISSF *vs.* INSPIRED2 | $2.33 \times 10^{-2}$ | $6.50 \times 10^{-3}$ | $6.70 \times 10^{-4}$ |
| SISSF *vs.* UniCRS | $2.83 \times 10^{-2}$ | $1.02 \times 10^{-2}$ | $6.70 \times 10^{-4}$ |

**Table 12 Hyperparameter space for the hyperparameter anaylsis on the ReDial dataset.**

| Learning rate | Temperature |
|---|---|
| 0.001 | 0.13 |
| 0.0001 | 0.11 |
| 0.00001 | 0.09 |
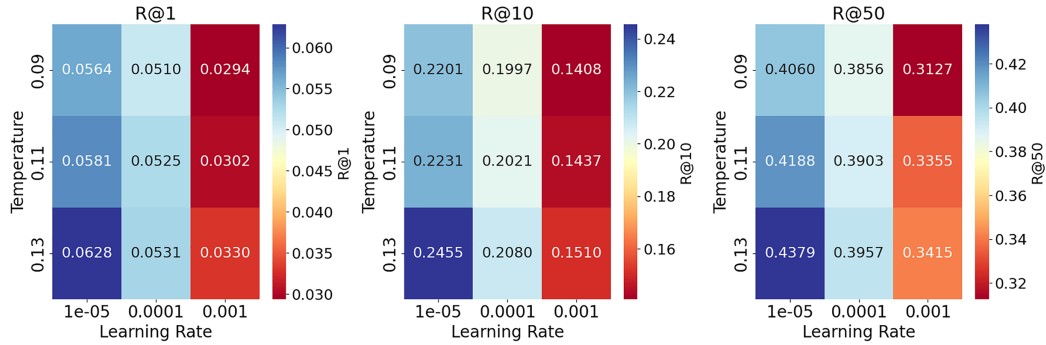

**Figure 5 Heatmaps of Recall (R@1, R@10, R@50) with different learning rate and temperature configurations on the ReDial dataset.** Lower learning rates combined with a temperature of 0.13 generally produce the highest recall values.

**Table 13 The results of hyperparameter analysis on the recommendation task on the ReDial dataset under different settings on Recall (R@1, R@10, R@50).**

| Dataset | ReDial | | | |
|---|---|---|---|---|
| Learning rate | Temperature | R@1 | R@10 | R@50 |
| 0.00001 | 0.13 | 0.0628 | 0.2455 | 0.4379 |
| 0.0001 | 0.13 | 0.0531 | 0.2080 | 0.3957 |
| 0.001 | 0.13 | 0.0330 | 0.1510 | 0.3415 |
| 0.00001 | 0.11 | 0.0581 | 0.2231 | 0.4188 |
| 0.0001 | 0.11 | 0.0525 | 0.2021 | 0.3903 |
| 0.001 | 0.11 | 0.0302 | 0.1437 | 0.3355 |
| 0.00001 | 0.09 | 0.0564 | 0.2201 | 0.4060 |
| 0.0001 | 0.09 | 0.0510 | 0.1997 | 0.3856 |
| 0.001 | 0.09 | 0.0294 | 0.1408 | 0.3127 |

## Hyperparameter analysis

During the experiment, we observed that the performance of our model demonstrated sensitivity to adjustments of specific hyperparameters. Consequently, we conducted a comprehensive hyperparameter analysis on the ReDial dataset (*Li et al., 2018*) to study the impact of the most influential hyperparameters on the performance of CRS's tasks and to find the optimal configuration for our model. Therefore, we chose two hyperparameters: the learning rate and the temperature of CL. Table 12 specifies the range of values for each hyperparameter. Following the grid search analysis (*Petro & Pavlo, 2019*), we have conducted exhaustive experiments on all combinations of the set of specified hyperparameters. This section presents a detailed report on the findings of our hyperparameter analysis.

### *Hyperparameter analysis on recommendation task*

The evaluation results of the hyperparameter analysis for all metrics of the recommendation task on the ReDial dataset (*Li et al., 2018*) are presented in Table 13.

**Table 14 Hyperparameter analysis for Distinct-2,3,4 on conversational task on the ReDial dataset with different learning rate and temperature values.**

| Dataset | ReDial | | | |
|---|---|---|---|---|
| Learning rate | Temperature | Distinct-2 | Distinct-3 | Distinct-4 |
| 0.00001 | 0.13 | 4.2239 | 5.5958 | 6.1557 |
| 0.0001 | 0.13 | 4.1145 | 5.5467 | 6.0147 |
| 0.001 | 0.13 | 2.9422 | 4.3277 | 4.9514 |
| 0.00001 | 0.11 | 4.1735 | 5.5614 | 6.1075 |
| 0.0001 | 0.11 | 3.8852 | 5.0548 | 5.7975 |
| 0.001 | 0.11 | 2.8974 | 4.1784 | 4.7458 |
| 0.00001 | 0.09 | 4.0031 | 5.5312 | 5.9857 |
| 0.0001 | 0.09 | 3.5234 | 4.1248 | 5.3625 |
| 0.001 | 0.09 | 2.7895 | 3.7536 | 4.0143 |

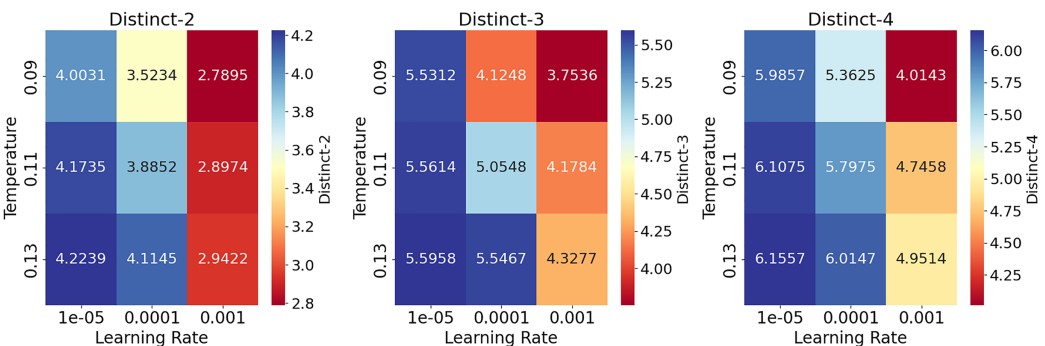

**Figure 6 Heatmaps to visualize the results of hyperparameter analysis on the conversational task on the ReDial dataset for Distinct-2,3,4 metrics under different settings.** The gradual decrease in the learning rate improves the results, with the best performance at a learning rate of 0.00001 and a temperature of 0.13.

Generally, lower learning rates produce better results on all metrics for higher temperature settings. Specifically, a temperature of (0.13) gives the best recall, especially for lower learning rates. The performance decreases slightly for high learning rates (0.0001), with a more significant drop for even higher learning rates (0.001). In summary, lower learning rates combined with the highest temperature yield the best recall performance, while higher learning rates across all temperate configurations significantly decrease performance. Figure 5 contains heatmaps for each recall metric, which provides a comprehensive visual representation of the data in Table 13.

### Hyperparameter analysis on conversational task

Table 14 shows the results of the hyperparameter analysis on the conversational task on the ReDial dataset (*Li et al., 2018*). Overall, higher learning rates tend to produce lower performance, which is improved gradually by decreasing the learning rate until the model reaches its best performance at (0.00001) for the range of values of temperature parameters. The lower temperatures also produce lower results than the highest value of

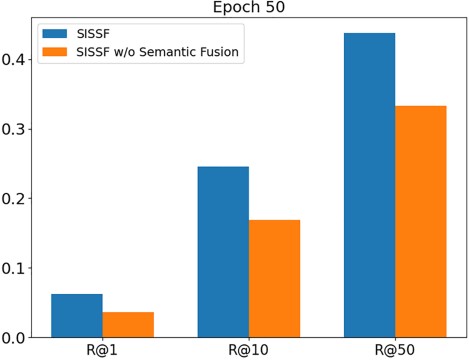
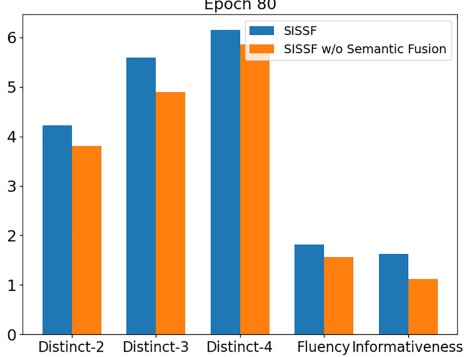

**Figure 7** Ablation study results on recommendation and conversational tasks on the ReDial dataset.

**Table 15 Results on ablation analysis for recommendation task on the ReDial dataset.** Comparison between SISSF and its variant SISSF w/o Semantic Fusion on all recall metrics at epoch 50.

| Dataset | ReDial | | |
|---|---|---|---|
| Model | R@1 | R@10 | R@50 |
| SISSF | 0.0628* | 0.2455* | 0.4379* |
| SISSF w/o Semantic fusion | 0.0365 | 0.1685 | 0.3331 |

**Note:**
Results marked with asterisk (*) are statistically significant according to the Wilcoxon rank sum test ($p$-value < 0.05).

(0.13). However, the values of all metrics are still comparatively high due to the usage of top-p sampling (*Holtzman et al., 2020*). Figure 6 provides heatmaps for each Distinct metric to visualize the data presented in Table 14.

## Ablation study

Based on the previous section, we conducted an ablation study on the ReDial dataset (*Li et al., 2018*) to measure the effectiveness of our method using the best hyperparameter configuration. Therefore, for ablation analysis, we incorporate a variant of our model: SISSF w/o Semantic Fusion, which removes semantic fusion between the conversational history and social information for recommendation and conversational tasks. Figure 7 visualized the result of the ablation study on both recommendation and conversational tasks on the ReDial dataset (*Li et al., 2018*). Next, we conducted an ablation study with the same variant to examine the impact of semantic fusion between social information and conversational history on personalized recommendations.

Furthermore, we performed an ablation study to compare the performance on the recommendation task using different graph methodologies. This analysis aimed to identify the best combination methods to represent the social relationship graph $H$ and the user-item interaction graph $T$ and assess the impact of the infer rating process on the recommendation task.

**Table 16 The results of the ablation analysis for the conversational task on the ReDial dataset, based on automatic and human evaluation metrics for SISSF with its variant at epoch 80.**

| Dataset | ReDial | | | | |
| --- | --- | --- | --- | --- | --- |
| Model | Distinct-2 | Distinct-3 | Distinct-4 | Fluency | Informativeness |
| SISSF | 4.2239* | 5.5958* | 6.1557* | 1.81* | 1.63* |
| SISSF w/o Semantic fusion | 3.8017 | 4.9015 | 5.8557 | 1.56 | 1.12 |

Note:
  Results marked with asterisk (*) are statistically significant according to the Wilcoxon rank sum test ($p$-value < 0.05).

**SISSF**

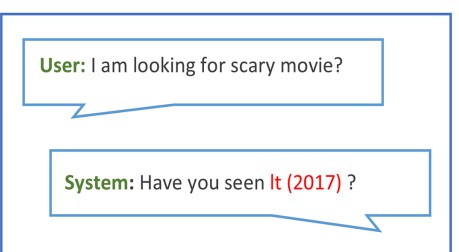

**SISSF w/o Semantic Fusion**

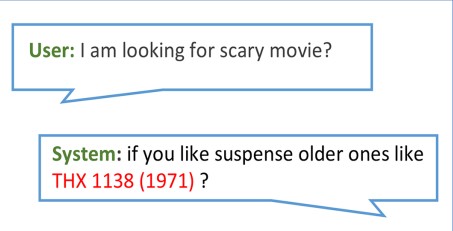

**Figure 8 Comparison of responses between SISSF and SISSF w/o Semantic Fusion on the ReDial dataset.**

### Ablation study on recommendation task

The ablation study results on the recommendation task on the ReDial dataset (*Li et al., 2018*) are summarized in Table 15. The results show that the operation of semantic fusion by applying CL between conversational history and social information is essential to improve the model's performance. Overall, the SISSF model outperforms SISSF w/o Semantic Fusion by a significant margin, which indicates the incorporation of semantic fusion leads to more accurate recommendations.

### Ablation study on conversational task

The ablation study results on the recommendation task on the ReDial dataset (*Li et al., 2018*) are shown in Table 16, which proves that semantic fusion contributes to the diversity of the response generation and its informativeness. SISSF performs superior in both automatic and human evaluations compared to its variant. The automatic evaluation shows comparatively close results due to the use of top-p sampling (*Holtzman et al., 2020*). We also conducted a human evaluation to better demonstrate the performance differences between the model and its variant. Furthermore, Fig. 8 shows examples of the responses generated by both models; SISSF demonstrates its ability to recommend contextually relevant items compared to SISSF w/o Semantic Fusion. For example, SISSF recommends a scary movie, It (2017), which is relevant to the user's query and a personalized recommendation, as this movie falls within the social circle of the seeker user. Conversely, SISSF without Semantic Fusion recommends a fiction movie that lacks personalization and contextual relevance.

**Table 17 Ablation study results on the impact of semantic fusion on personalized recommendations on the ReDial dataset.** The table compares the performance of the SISSF model to its variant SISSF w/o Semantic Fusion based on the R@1 metric at epoch 50.

| Dataset | ReDial |
|---|---|
| Model | R@1 |
| SISSF | 0.1132* |
| SISSF w/o Semantic Fusion | 0.0154 |

Note:
Results marked with asterisk (*) are statistically significant according to the Wilcoxon rank sum test (*p*-value < 0.05).

**Table 18 Ablation study on the recommendation task showing Recall@1, Recall@10, and Recall@50 for different combinations of graph technologies.**

| Dataset | ReDial | | | |
|---|---|---|---|---|
| Social relationship graph | User-item interaction graph | R@1 | R@10 | R@50 |
| GCN | GCN | 0.0440 | 0.1942 | 0.3801 |
| GTA | GTA | 0.0493 | 0.2010 | 0.3918 |
| R-GCN | GTO | 0.0503 | 0.2021 | 0.4014 |
| GTO | GTO | 0.0511 | 0.2101 | 0.4065 |
| R-GCN | R-GCN | 0.0541 | 0.2133 | 0.4200 |
| GTO | R-GCN | 0.0628* | 0.2455* | 0.4379* |

Note:
Results marked with asterisk (*) are statistically significant according to the Wilcoxon rank sum test (*p*-value < 0.05).

## Ablation study on the impact of semantic fusion on personalized recommendations

The primary purpose of this ablation study is to highlight the reliability of extracting social information from conversational history for personalized recommendations. To achieve this, first, we obtain the items that the users have not yet interacted with but are present within their social circle's interests. Next, the recommendation task was evaluated on the ReDial dataset (*Li et al., 2018*) by testing whether the first item recommended by the model belongs to the set of those items for the seeker users. Table 17 indicates SISSF outperforms SISSF w/o Semantic Fusion, achieving a performance score of 0.1132 compared to 0.0154 in R@1. These findings demonstrate that extracting social information and applying semantic fusion to integrate it with conversational history is not merely a method of overfitting the dataset.

## Ablation study on the impact of graph architectures and infer rating process on recommendation performance

In this ablation study, we investigated the use of different graph types to identify the best models for the user-user social graph $H$ and the user-item interaction graph $T$ on the ReDial dataset (*Li et al., 2018*). Table 18 shows that employing GTO (*Shi et al., 2021*) for H and R-GCN (*Schlichtkrull et al., 2018*) for $T$ yields the best results. Moreover, using graph methodologies such as GTO (*Shi et al., 2021*), GCN (*Kipf & Welling, 2017*), or GAT (*Veličković et al., 2018*) for $T$ did not contribute significantly to improving the outcome.

This indicates leveraging NGCF (*Wang et al., 2019*) to infer ratings, which is used to weigh the relationships among nodes within R-GCN (*Schlichtkrull et al., 2018*), enhances the performance. These findings underscore the importance of incorporating inferred ratings with NGCF (*Wang et al., 2019*) to optimize the model's performance. The reason behind the improvement of H with GTO (*Shi et al., 2021*) over other models is due to its ability to effectively weight relationships between nodes (*Brody, Alon & Yahav, 2022*).

## CONCLUSION AND FUTURE WORK

In conclusion, SISSF addresses the fundamental challenge of incorporating social dynamics into conversational recommendations. Our approach demonstrates a principled method for extracting and utilizing social information from conversational datasets, even when explicit social structures are unavailable. Extensive evaluation of two widely recognized public datasets has yielded several significant findings. The results consistently demonstrate the superiority of SISSF over existing baseline approaches across multiple performance metrics in both recommendation and conversational tasks. The implications of this work extend beyond immediate performance improvements. Our framework establishes a new paradigm for understanding and utilizing social dynamics in CRS. The success of our approach in extracting and utilizing implicit social information highlights its potential for applications in domains where explicit social structures are either absent or incomplete.

Despite its strong performance, SISSF faces certain limitations. One major challenge is the cold start problem, where generating recommendations for new users or items is difficult due to a lack of prior interactions. Additionally, deploying SISSF in real-world applications presents challenges, particularly maintaining an operational online environment where interactions and relationships must be continuously updated. These limitations open avenues for future work, particularly in developing cross-domain approaches that leverage social information from one domain to enrich recommendations in another, effectively mitigating data sparsity and enhancing system robustness.

### Funding
This research was funded by the Researchers Supporting Project number (RSP2024R476), King Saud University, Riyadh, Saudi Arabia. The funders had no role in study design, data collection and analysis, decision to publish, or preparation of the manuscript.

### Grant Disclosures
The following grant information was disclosed by the authors:
King Saud University, Riyadh, Saudi Arabia: RSP2024R476.

### Competing Interests
The authors declare that they have no competing interests.

## Author Contributions

- Abdulaziz Mohammed conceived and designed the experiments, performed the experiments, performed the computation work, authored or reviewed drafts of the article, and approved the final draft.
- Mingwei Zhang analyzed the data, prepared figures and/or tables, authored or reviewed drafts of the article, and approved the final draft.
- Gehad Abdullah Amran analyzed the data, prepared figures and/or tables, authored or reviewed drafts of the article, and approved the final draft.
- Husam M. Alawadh analyzed the data, prepared figures and/or tables, authored or reviewed drafts of the article, and approved the final draft.
- Ruizhe Wang analyzed the data, prepared figures and/or tables, and approved the final draft.
- Amerah Alabrah analyzed the data, prepared figures and/or tables, authored or reviewed drafts of the article, and approved the final draft.
- Ali A. Al-Bakhrani analyzed the data, prepared figures and/or tables, authored or reviewed drafts of the article, and approved the final draft.

## Data Availability

The original ReDial dataset is available at https://redialdata.github.io/website.

The original INSPIRED dataset is available at GitHub: https://github.com/sweetpeach/Inspired.

The project code and the code for preprocessing

the datasets used by our project are available at Zenodo: Aziz. (2025). glad47/SISSF: final version (v1.0.0). Zenodo. https://doi.org/10.5281/zenodo.15368530.

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
