# Peer review of "A social information sensitive model for conversational recommender systems"

_PeerJ Computer Science, doi:10.7717/peerj-cs.3067_

## Round 0.1 · original submission · Major Revisions

Dear Authors,

The reviewers think that your manuscript requires significant revisions, particularly in readability, statistical analysis, simplification, better justification the followed methodology, and usage of stronger baselines. We do encourage you to address the concerns and criticisms of the reviewers with respect to reporting, experimental design, and validity of the findings and resubmit your article once you have updated it accordingly. Following should also be addressed:

1. Detailed comparative and statistical analyses should be performed.
2. Please pay special attention on the usage of abbreviations. Spell out the full term at its first mention, indicate its abbreviation in parenthesis and use the abbreviation from then on.
3. Equations should be used with correct equation number. Please do not use “as follows”, “given as”, etc. Explanation of the equations should also be checked. All variables should be written in italic as in the equations. Their definitions and boundaries should be defined. Necessary references should be provided.
4. Many of the equations are part of the related sentences. Attention is needed for correct sentence formation.

Best wishes,

**Language Note:** The review process has identified that the English language must be improved. PeerJ can provide language editing services - please contact us at [email protected] for pricing (be sure to provide your manuscript number and title). Alternatively, you should make your own arrangements to improve the language quality and provide details in your response letter. – PeerJ Staff

·

Basic reporting

This paper investigates the conversational recommendation task and introduces a novel approach that integrates social information as additional context to improve the accuracy of conversational recommendations and the naturalness of conversational utterances.

Overall, this manuscript is well structured and the coverage of related works to the proposed approach is sufficient and systematic.

However, this manuscript often includes grammar errors and incomplete sentences, which need additional improvements to ensure readability. Detailed comments on the recognised writing issues will be provided in the later "additional comments".

Regarding the motivation of this work, there are some arguable aspects:

1. The introduction mentions that "Results show (Zhou et al., 2020a, 2022; Pugazhenthi and Liang, 2022) using only seq2seq to implement the CRS (Christensen et al., 2018) reveals improvement opportunities where external data sources could be integrated with the conversational data through semantic fusion." However, this can be a biased summary of existing studies, since the use of the seq2seq model is not the most promising solution in the literature, and this discussion can be misleading to the reader about the current research agreement in the CRS community.

2. Regarding one of the main arguments in this work: "... This absence limits the system’s capacity to capture and utilize valuable social dynamics in the recommendation process. Second, there exists a substantial semantic gap between social information and conversation history, making it challenging to effectively integrate user preferences expressed in natural language with patterns derived from social relationships". It is good to highlight the value of social information in the development of conversational recommendation techniques. However, it is necessary to highlight how the social information can be reliably extracted from the conversational histories. Otherwise, the extraction of social information can be a way of overfitting the existing dataset and harming the generalizability of the resulting model.

The description of the methodology can be improved, especially with the use of many variables. Some specific points: (1) the variable 'C' has been used to indicate both conversation in line 215 and a shared set of items C(ij) in line 219. (2) "Let + Avr(C(i j)), be the largest integer less than or equal to the average value of all ratings of both users ui and u j for items in C(i j), Let K * R n×n define the matrix of user relationships where k i j = + Avr(C(i j)), if there is a social relationship between u j and u i, and k ij= 0 otherwise." The definition of user relationships from line 219 to line 223 is ambiguous. We need to know how exactly "if there is a social relationship between u_j and u_i", namely how can we determine there is a social relationship between two users. (3) In line 242, "we construct one paragraph P ...". However, how paragraph P is constructed is not clearly described. A reasonable guess is that this paragraph comes from historical conversations, but is not explicitly stated. (4) From line 286: "Thus, we perform contrastive learning for each conversation as follows: (e_c, e_s).". It is not confusing how two variables can enable performing contrastive learning. In addition, in lines 287 and 288, it is first stated that "positive examples involve data representations directly related to the same conversation, while negative examples represent other data representations in batch." Then, in lines 292 and 293, it is again stated that "where h and h + are two types of data representations for a conversation, one of which is the data representation for the conversation history, and the other represents the social information.". Not only are the variables not consistently used, but the definition of positive and negative parts of the input to contrastive learning is unclear.

Writing consistency can also be improved. The first half of this work uses "user" while the second half of the manuscript starts using "seeker".

Experimental design

To evaluate the performance of the proposed approach, this work conducts experiments on two widely used CRS datasets and compares the model's performance to a list of baselines. The experimental results show that the proposed approach can outperform the baselines consistently over two datasets and conducted a rich set of ablation studies to justify the contribution of the social information inclusion component.

However, regarding the reliability of the experimental design, there are some issues that need to be addressed:

1. The construction of the social information is unclear. By assuming that the way to calculate the social relationships between users is detailed in the datasets subsection of the experimental setup section, the exact way to implement the estimation of social connections between users is not clearly stated. For example, in lines 394 to 395, "we manually annotate each movie mentioned in conversations to add users’ movie interaction information to the INSPIRED dataset". The annotation process is not clearly described in the text, which can hinder the reproducibility of this work. In addition, the corresponding paragraph of such a sentence is not easy to follow; adding some illustrative examples can be helpful.

2. The baseline approaches that have been included in this work are rather outdated; the latest one is from C2CRS. The comparison to recent baselines can further justify the timely contribution of this work.

3. This work conducts ablation studies to show the value of the inclusion of the semantic fusion component. It can be further improved if some recommended examples can be provided to explicitly illustrate the benefits.

Validity of the findings

The overall findings are properly grounded in the experimental results.

Additional comments

Some additional comments on the writing issues:

1. Lines 82-83: "Followed by concatenating both profiles and performing mean pooling to obtain enhanced user representations." This sentence is incomplete.

2. Lines 222-223: "which represents the user-user social graph H, Let J(i) be the set of users whom u i directly connected." This sentence is not properly positioned and is incomplete.

3. Lines 225: "to 1) Generate appropriate responses s t+1 and 2) Recommends items", should be "recommend personalised items".

4. Lines 461 - 462: "We follow Algorithm 1 with PyTorch 1 and WSDM2022-C2CRS (Zhou et al., 2022) 2 to implement our approach." This sentence is confusing, and algorithm 1 was not introduced explicitly in the text previously.

Reviewer 2 ·

Basic reporting

The paper's abstract can be improved by a native English speaker for better presentation of the paper. However, other parts are well-written with acceptable structure.

Experimental design

The submission clearly defined the research question, and it is relevant.

Validity of the findings

I think the result analysis and discussion can be improved with additional statistical analysis. I recommend that the author use R Commander in RStudio for that.

Cite this review as

Reviewer 3 ·

Basic reporting

The manuscript fails to meet the basic reporting standards in multiple areas, including clarity, literature review, and structure.

- Clarity & Language: The writing is overly complex, making it difficult to follow. Many sentences are bloated with redundant phrases. For example, in the introduction, the sentence: "Conversational Recommender Systems (CRS) have emerged as a significant advancement in recommendation technology, facilitating natural language interactions for more effective item suggestions." could be rewritten as: "Conversational Recommender Systems (CRS) enhance item recommendations through natural language interactions." This excessive verbosity persists throughout the manuscript.

- Literature References & Context: The literature review is a mere listing of prior works without critical evaluation. In Section: Related Work, the authors write: "Recommender systems are essential in providing personalized suggestions within commercial platforms, employing a variety of methodologies to cater to user preferences." Yet, they do not clearly justify how their approach is different or superior. The discussion of previous models lacks any meaningful critique.

- Figures & Tables: Figures are included but lack clear explanations. For instance, in Figure 1, there is no in-depth discussion on why their integration of social information is novel compared to existing work.

- Self-contained & Relevant Results: The paper claims improvements but lacks proper justification. The results section makes broad claims such as: "Our SISSF framework demonstrated significant improvements over baseline models across all metrics." Yet, the reported performance improvements (e.g., Recall@1 increase of only 0.001 over the best baseline) do not justify the term "significant" without a proper statistical analysis.

Improvements Needed:

- Simplify the writing to improve readability.
- Critically evaluate prior work instead of just listing references.
- Justify the figures with stronger explanations.
- Avoid overstatements regarding the significance of results.

Experimental design

The manuscript lacks methodological rigor and fails to provide sufficient details for reproducibility.

- Research Question: The research problem is not clearly articulated. The paper states: "We introduce a social information-sensitive semantic fusion approach that employs contrastive learning (CL) to bridge the semantic gap between generated social information and conversation history." However, it does not clearly explain why contrastive learning is necessary or how it addresses an existing limitation in CRS.

- Methodological Complexity Without Justification: The model introduces unnecessary complexity. For example, in the Encoding of User-User Relationship Graph, it states: "To ensure the model can accurately encode the significance of each relationship between two users, we use R-GCN (Schlichtkrull et al., 2018)." There is no justification for why R-GCN is chosen over simpler alternatives, nor is there an ablation study comparing different graph-based approaches.

- Reproducibility Issues: The methodology lacks critical details, especially in the experimental setup. The dataset pre-processing is vaguely described, and key hyperparameters are not provided. Without sufficient replication details, this study cannot be independently validated.

Improvements Needed:

- Clearly define the research problem and knowledge gap.
- Justify why each methodological choice is necessary.
- Provide detailed hyperparameters and preprocessing steps to ensure reproducibility.

Validity of the findings

The validity of the findings is questionable due to weak baseline comparisons, overstatements, and insufficient statistical analysis.

- Weak Baseline Comparisons: The paper ignores more recent CRS models that use reinforcement learning. It compares against older baselines (e.g., KGSF, ReDial) but does not consider newer state-of-the-art methods. This weakens the validity of its claims.

- Overstatement of Results: In the Evaluation on Recommendation Task, the authors claim: "Our model significantly outperforms C2-CRS." Yet, the actual Recall@1 improvement is only 0.001, which is not statistically significant. The lack of statistical tests (e.g., t-tests or confidence intervals) raises doubts about the robustness of the results.

- Failure to Discuss Limitations: The paper ignores potential weaknesses of the approach. For example, the reliance on contrastive learning assumes that social information is always beneficial, but no analysis is provided on cases where social information might mislead recommendations.

Improvements Needed:

- Include stronger baselines from recent CRS literature.
- Perform statistical significance tests before claiming superiority.
- Discuss the limitations of the approach.

Additional comments

- Disorganized Structure: Sections lack logical transitions, making the paper difficult to follow. The jump from Encoding Multi-form Data to Applying Contrastive Learning is abrupt, with no clear connection.

- Unsubstantiated Claims: Statements such as: "These findings suggest that incorporating social context through CL can significantly improve the quality and relevance of recommendations in conversational systems." are not backed by any detailed causal analysis.

- Poorly Defined Contributions: The contributions section merely restates the methodology instead of clearly outlining what is novel.

Improvements Needed:

- Improve section transitions to enhance readability.
- Remove overly broad claims without evidence.
- Clearly define the paper’s novel contributions.

Cite this review as

Reviewer 4 ·

Basic reporting

The paper is generally well-written, with a clear abstract and structured organization. Figures are mostly clear and relevant and support the methodology (e.g., Fig. 2 provides a helpful visual on semantic fusion). The introduction is informative and sets up the problem well, situating it within the current Conversational Recommender Systems (CRS) landscape. Public datasets (ReDial and INSPIRED) are appropriate and align with best reproducibility practices. References are thorough and current and include both foundational and recent works.

The following areas of concern require further explanation and corrections in the paper.
a. Some sentences are overly complex or awkwardly phrased; a language polish would help improve clarity (e.g., Lines 56–64).

b. Figure 1 is somewhat ambiguous and could be redrawn to better highlight the contrast between traditional CRS and the proposed SISSF approach. Figure 1 must be redrawn to clearly show the distinction between SISSFCRS and traditional CRS.

c. The definition of “liked” items in the user-user graph is vague (Line 220) — a clear threshold for “liking” (e.g., rating ≥ 3) should be explicitly stated.

d. Consistency in notation is needed (e.g., “+” before Avr is unconventional and confusing).

Experimental design

The research question is well-defined and relevant: integrating social information through contrastive learning to improve CRS performance.

There are a few concerns here:
a. When constructing the user-interaction bipartite graph, how did the authors identify the opinion of user u_i on item i_j based on the rating r_ij? Please elaborate on what the authors meant by opinion in this case. Consequently, when constructing the user-user social homogeneous graph, a relationship between users is established if they share at least one liked item. Please define “like”. Were there any assumptions made about the ratings that indicate the ‘like’?

b. The definition of [Avr(C(i j))] may require elaboration.

c. Some methodological steps could benefit from more concrete explanations. For instance:
- How were negative samples selected during contrastive learning?
- Was there any sampling strategy to balance user roles (seeker/recommender)?

d. While Algorithm 1 is useful, more pseudocode-like structures or flowcharts could improve accessibility to readers unfamiliar with this integration pipeline.

e. Although human evaluations were conducted, more details of the annotation process and procedure are necessary.

Validity of the findings

The validity of the findings is supported by a comprehensive set of evaluations, including both automatic metrics (e.g., Recall@K, Distinct-n) and human assessments (fluency and informativeness). The results consistently show that the SISSF model outperforms a strong set of baselines, including recent CRS systems like KGSF and C2-CRS.

There are a few concerns here:
a. While gains show the improvement in R@1 (from 0.053 to 0.054) on ReDial is minimal, this should be discussed with more nuance.

b. Potential overfitting risks or limitations (e.g., scalability, real-world deployment issues) are not sufficiently discussed.

c. In typical recommendation task evaluation, researchers mostly focus on the precision and NDCG measures. Give reasons why such measures are not considered in this experiment.

Cite this review as

---

## Round 0.2 · Minor Revisions

Dear Authors,

Thank you for improving the paper according to the suggestions. It is recommended that the minor concerns and criticisms raised by Reviewer 2 are addressed, and that the paper is resubmitted once the necessary updates have been made.

Best wishes,

Reviewer 2 ·

Basic reporting

The paper presentation is clear and the literature references are sufficient, but the paper needs a pseudocode for the proposed method and a statistical analysis for the results obtained (I recommend to use Rcommander in Rstudio for that {p-values, statistical summary...etc.)

Experimental design

no comment

Validity of the findings

the paper needs a pseudocode for the proposed method and a statistical analysis for the results obtained (I recommend to use Rcommander in Rstudio for that {p-values, statistical summary...etc.)

Additional comments

The paper presentation is clear and the literature references are sufficient, but the paper needs a pseudocode for the proposed method and a statistical analysis for the results obtained (I recommend to use Rcommander in Rstudio for that {p-values, statistical summary...etc.)

Cite this review as

Reviewer 5 ·

Basic reporting

- The authors have addressed prior concerns regarding grammatical inconsistencies, verbosity, and incomplete sentences.
- The ambiguous phrases have been rewritten, resulting in a more concise and readable text throughout the manuscript.
- The structure of the paper adheres to conventional academic standards, with clearly delineated sections including Introduction, Related Work, Methodology, Experiments, and Conclusion.
- The figures and tables are now supported by more informative captions and are referenced appropriately within the text.
Overall, the paper has been corrected according to comments of editor and reviewers.

Experimental design

-The authors have explained the datasets (ReDial and INSPIRED), including how social relationships and item interactions were extracted and annotated. The annotation process is documented with illustrative examples to support reproducibility.
-Hyperparameters and implementation details have been explicitly listed in the revised manuscript.
-Additionally, the paper presents both automatic and human evaluation metrics for performance assessment across two benchmark datasets.

Validity of the findings

- The findings presented in the manuscript are supported by the experimental results.
- The authors have given the statistical analysis by reporting p-values and highlighting statistically significant improvements over baseline models.
-The authors have demonstrated the generalizability of their approach through consistent improvements across multiple datasets.

Cite this review as

---

## Round 0.3 · accepted · Accept

Dear Authors,

Thank you for addressing the reviewers' comments. Your manuscript now seems sufficiently improved and ready for publication.

Best wishes,


Reviewer 2 ·

Basic reporting

The paper is well-written with good structure and Self-contained with relevant results to hypotheses.

Experimental design

The research experiments are original and met the scope and aims of the journal.

Validity of the findings

The findings are novel and they adding the statistical meaning of the results obtained

Additional comments

I am satisfied with current version.

Cite this review as

Reviewer 4 ·

Basic reporting

As my previous comments the paper is well organized.

Experimental design

The authors have addressed all my previous concerns.

Validity of the findings

The authors have addressed all my previous concerns.

Additional comments

No comments.

Cite this review as